# Hierarchical Rectified Flow Matching with Mini-Batch Couplings

## Abstract

Flow matching has emerged as a compelling generative modeling approach that is widely used across domains. During training, flow matching learns to model a velocity field. At inference, to generate samples, an ordinary differential equation (ODE) is numerically solved via forward integration of the modeled velocity field. To better capture the multi-modality that is inherent in typical velocity fields, hierarchical flow matching was recently introduced. It uses a hierarchy of ODEs that are numerically integrated when generating data. Each level of the hierarchy of ODEs captures the distribution of the next level, just like vanilla flow matching uses the velocity field to capture a multi-modal data distribution. While this hierarchy enables to model multi-modal distributions at any hierarchy level, the complexity of the modeled distributions remains identical across levels of the hierarchy. In this paper, we study how to gradually adjust the complexity of the distributions across different levels of the hierarchy via mini-batch couplings. We show the benefits of mini-batch couplings in hierarchical rectified flow matching via compelling results on synthetic and imaging data.

## 1 Introduction

Flow matching (Lipman et al., 2023; Liu et al., 2023a; Albergo & Vanden-Eijnden, 2023) has gained significant attention across computer vision (Esser et al., 2024; Liu et al., 2023b), robotics (Zhang & Gienger, 2024), computational biology (Yim et al., 2023; Jing et al., 2023), and time series analysis (Chen et al., 2024; Zhang et al., 2024). This is largely due to its ability to generate high-quality data and due to its simple simulation-free learning of a data distribution. For this, it uses 1) an intermediate state, which is computed by (linearly) interpolating between a sample from a known source distribution and a randomly drawn data point, and 2) the velocity at this intermediate state. This velocity controls a neural ordinary differential equation (ODE), which governs the transformation of the samples from the source distribution to the target data distribution. Note, the distribution of velocities at an intermediate state is multimodal (Zhang et al., 2025).

In classic flow matching, velocities at interpolated states are modeled via a parametric deep net using a mean squared error (MSE) objective. It is known that the MSE objective used in classic flow matching does not permit to capture the multimodal velocity distribution. Instead, training in classic flow matching leads to a velocity model that captures the mean of the velocity distribution. Capturing the mean of the velocity distribution is sufficient for characterizing a multimodal data distribution (Liu et al., 2023a). However, it inevitably results in curved forward integration paths, making the sampling process inefficient. Recently, hierarchical flow matching (Zhang et al., 2025) was suggested as an approach to model the multimodal velocity field via coupled ODEs.

To model the multimodal velocity field, hierarchical flow matching essentially applies a flow matching formulation in the velocity space by matching 'acceleration.' It was also suggested to expand the idea further towards an arbitrary hierarchy level. While this enables to model multimodal velocity distributions, and also distributions at arbitrary hierarchy levels, the complexity of the modeled distributions remains identical across all levels of the hierarchy. Said differently, the velocity distribution that hierarchical flow matching models across levels of its hierarchy is no easier than the original data distribution, potentially limiting benefits.

We hence wonder: *can we gradually simplify the complexity of the distributions across hierarchy levels?* For simplicity, in this paper, we focus on two hierarchy levels. Interestingly, we find mini-

batch couplings to provide a compelling way to control the "ground-truth" velocity distribution. Instead of computing intermediate states by interpolating between samples independently drawn from both the known source distribution and the dataset, we draw a mini-batch of samples from both the source distribution and the dataset, and subsequently couple them, e.g., via a procedure like optimal transport. Intuitively, considering as an extreme situation a mini-batch containing the entire dataset leads to a velocity distribution that is unimodal.

Empirically, we find that hierarchical flow matching with mini-batch coupling in the data space consistently improves the generation quality of vanilla hierarchical flow matching and vanilla flow matching with optimal transport coupling. Jointly coupling mini-batch samples in data and velocity space leads to further benefits if the number of neural function evaluations (NFEs) is low.

## 2 PRELIMINARIES

**Rectified Flow (RF).** A rectified flow models an unknown target data distribution $\rho_1$ given a dataset $\mathcal{D} = \{x_1\}$, where we assume data points $x_1 \sim \rho_1$. Given a known source distribution $\rho_0$ (e.g., standard Gaussian), at inference time, source samples $x_0 \sim \rho_0$ evolve from time $t = 0$ to time $t = 1$ following the ordinary differential equation (ODE)

$$dz_t = v(z_t, t)dt, \text{ with } z_0 \sim \rho_0, \quad t \in [0, 1]. \tag{1}$$

Here, $v(z_t, t)$ is a velocity field that depends on time $t$ and the current intermediate state $z_t$. This ODE-based sampling enables to capture multimodal data distributions.

At training time, flow matching learns the velocity field $v(z_t, t)$ by minimizing the $\ell_2$-loss between the predicted velocity $v(x_t, t)$ and a ground-truth velocity $v_{\text{gt}}(x_t, t)$. To obtain the ground-truth velocity we first define an intermediate state $x_t$ which, in a rectified flow formulation, is obtained by linearly interpolating between a randomly drawn source sample $x_0$ and a randomly drawn data point $x_1$, i.e.,

$$x_t = (1 - t)x_0 + tx_1, \quad \text{where } x_0 \sim \rho_0, \, x_1 \sim \mathcal{D}. \tag{2}$$

Interpreting the intermediate state $x_t$ as a location, we obtain the ground-truth velocity $v_{\text{gt}}(x_t, t) = \partial x_t / \partial t = x_1 - x_0$. Combined, training addresses

$$\inf_v \mathbb{E}_{x_0 \sim \rho_0, x_1 \sim \mathcal{D}, t \sim U[0,1]} \left[ \|x_1 - x_0 - v(x_t, t)\|_2^2 \right], \tag{3}$$

where the infimum is over all measurable velocity fields. In practice, $v(x_t, t)$ is parameterized by a deep net with trainable parameters $\theta$, i.e., $v(x_t, t) \approx v_\theta(x_t, t)$. The optimization minimizes over $\theta$.

However, for a given $t$ and $x_t$, different pairs $(x_0, x_1)$ will yield different ground-truth velocities. The ground-truth velocity distribution at a given time $t$ and intermediate state $x_t$ is hence multimodal. However, the $\ell_2$-loss averages these velocities, resulting in the 'optimal' velocity field: $v^*(x_t, t) = \mathbb{E}_{\{(x_0, x_1, t):(1-t)x_0 + tx_1 = x_t\}}[v_{\text{gt}}(x_t, t)]$. According to Theorem 3.3 by Liu et al. (2023a), using $v^*$ in Equation (1) ensures that the stochastic process has marginal distributions consistent with the linear interpolation in Equation (2).

To capture multimodal velocity distributions, hierarchical flow matching (Zhang et al., 2025) was introduced. It explicitly models the multimodal velocity distributions at each time $t$ and intermediate state $x_t$, enabling a more expressive generative framework.

**Hierarchical Rectified Flow (HRF).** To model the "ground-truth" velocity distribution more accurately, hierarchical rectified flow extends the classic rectified flow framework by focusing on velocities rather than locations. This approach effectively involves learning acceleration. In a classic rectified flow, the time-dependent location $x_t$ is computed from pairs $(x_0, x_1)$, and the ground-truth velocity $v_{\text{gt}}(x_t, t) = \partial x_t / \partial t$ is used to train a velocity model $v_\theta(x_t, t)$.

In hierarchical rectified flow, a source velocity sample $v_0 \sim \pi_0$ is drawn from a known velocity distribution $\pi_0$, while a target velocity sample $v_1(x_t, t) \sim \pi_1(v; x_t, t)$ is defined at each time $t$ and location $x_t = (1 - t)x_0 + tx_1$. For rectified flow, $v_1(x_t, t)$ is computed via $x_1 - x_0$, and these samples follow the ground-truth velocity distribution $\pi_1(v; x_t, t)$.

To learn acceleration, a new time axis $\tau \in [0, 1]$ is introduced, and a time-dependent velocity $v_\tau(x_t, t) = (1 - \tau)v_0 + \tau v_1(x_t, t)$ is constructed. The ground-truth acceleration is then obtained as

$a(x_t, t, v_\tau, \tau) = \partial v_\tau / \partial \tau = v_1(x_t, t) - v_0 = x_1 - x_0 - v_0$. For a fixed $(x_t, t)$, this leads to the ODE:

$$du_\tau(x_t, t) = a(x_t, t, u_\tau, \tau)d\tau, \quad \text{with } u_0 \sim \pi_0. \tag{4}$$

Here, $a(x_t, t, u_\tau, \tau)$ is the expected acceleration vector field: $a(x_t, t, u_\tau, \tau) = \mathbb{E}_{\pi_0, \pi_1(v; x_t, t)}[V_1 - V_0 | V_\tau = u]$. The acceleration vector field is learned by addressing

$$\inf_a \mathbb{E}_{x_0 \sim \rho_0, x_1 \sim \mathcal{D}, t \sim U[0,1], v_0 \sim \pi_0, \tau \sim U[0,1]} \left[ \|(x_1 - x_0 - v_0) - a(x_t, t, v_\tau, \tau)\|_2^2 \right]. \tag{5}$$

In practice, the acceleration is parameterized via a deep net $a_\theta(x_t, t, v_\tau, \tau)$, and the model is trained by minimizing this objective over the parameters $\theta$.

During sampling, coupled ODEs are used:

$$\begin{cases} du_\tau(z_t, t) = a(z_t, t, u_\tau, \tau)d\tau, & u_0 \sim \pi_0, \ \tau \in [0, 1], \\ dz_t = u_1(z_t, t)dt, & z_0 \sim \rho_0, \ t \in [0, 1]. \end{cases} \tag{6}$$

These ODEs map $z_0 \sim \rho_0$ to $z_1 \sim \rho_1$. Sampling involves drawing $v_0 \sim \pi_0$ and $x_0 \sim \rho_0$, integrating forward to obtain $v_1(x_0, 0)$, and then performing location updates iteratively until reaching $x_1$. This procedure can be implemented using the vanilla Euler method and the trained $a_\theta$.

Considering the training objective for acceleration matching (Equation (5)) and the coupled ODEs for sampling (Equation (6)), both can be naturally extended to any depth. In this paper, we focus solely on depth-two HRF (HRF2) models.

**Minibatch Optimal Transport.** Optimal Transport (OT) seeks to find an optimal coupling of two distributions that minimizes an expected transport cost (Villani, 2009). Suppose $\alpha$ and $\beta$ are two distributions in $\mathbb{R}^d$, and $c : \mathbb{R}^d \times \mathbb{R}^d \to \mathbb{R}$ is some distance. Then OT aims to find the solution of the following optimization problem:

$$\inf_{\gamma \in \Gamma} \int_{\mathbf{R}^d \times \mathbf{R}^d} c^2(x, y) d\gamma(x, y), \tag{7}$$

where $\Gamma$ is the set of all joint distributions with marginals $\alpha$ and $\beta$. When $\alpha$ and $\beta$ are both empirical distributions, OT reduces to linear programming, which is computationally expensive when the data size is large (Peyré et al., 2019). While OT is computationally expensive for large datasets, mini-batch OT (Fatras et al., 2020; 2021) was introduced as an alternative: a small batch of the data is used to calculate the coupling, obtaining an unbiased estimator of the underlying truth (Fatras et al., 2020). Although mini-batch OT incurs an error compared to the exact OT, it has found use in practice (Deshpande et al., 2018; 2019; Pooladian et al., 2023; Tong et al., 2024; Cheng & Schwing, 2025). Tong et al. (2024); Pooladian et al. (2023) showed that training and inference are more efficient with mini-batch OT.

## 3 APPROACH

In Section 3.1, we use 1D data to illustrate how mini-batch couplings in data space and velocity space affect the velocity distribution and the generation of velocity samples. This motivates the development of HRF with mini-batch coupling and extension of its theory. In Section 3.2, we introduce the training of HRF2 with mini-batch coupled data points. In Section 3.3, we explain how mini-batch coupling for velocity is achieved by leveraging a pre-trained model. In Section 3.4, we introduce a two-stage approach that combines mini-batch data coupling and velocity coupling.

### 3.1 VELOCITY DISTRIBUTION

For vanilla HRF2, the source and target distributions are independent, meaning $\gamma(x_0, x_1) = \rho_0(x_0)\rho_1(x_1)$. Consequently, as derived by Zhang et al. (2025), at time $t = 0$, the velocity distribution becomes $\pi_1(v; x_t, t) = \rho_1(x_t + v)$, making it a shifted version of the data distribution. Hence, learning this distribution is as challenging as directly modeling the data distribution.

To control the complexity of the velocity distribution, we study couplings in data space and velocity space. Concretely, for couplings in data space, we sample from a distribution $\gamma(x_0, x_1)$, which doesn't factorize. This can be achieved by coupling samples within each mini-batch, i.e., samples

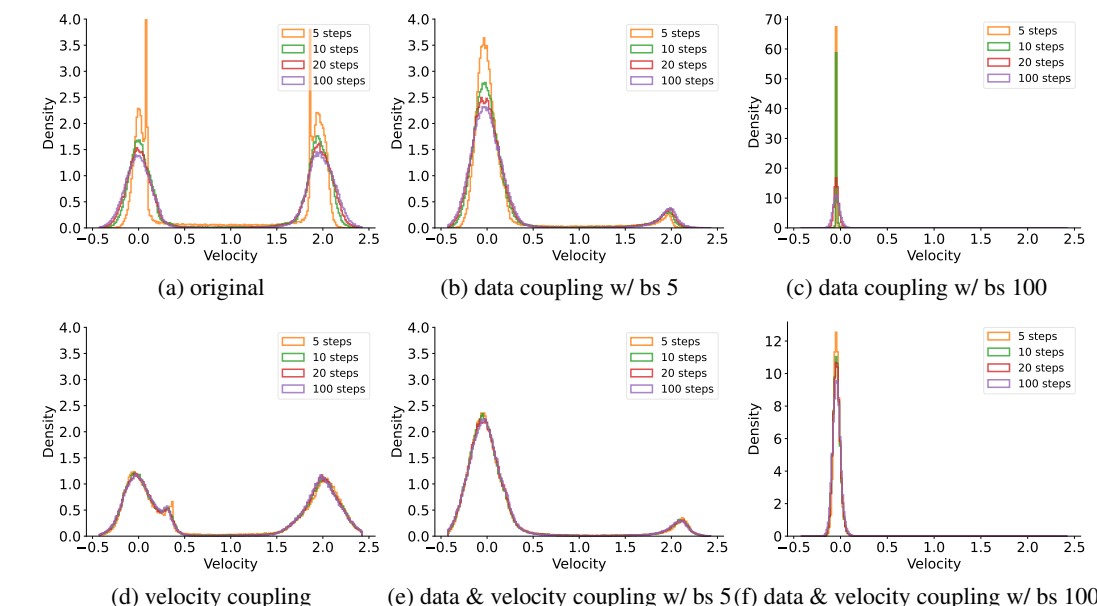

(a) original  (b) data coupling w/ bs 5  (c) data coupling w/ bs 100

(d) velocity coupling  (e) data & velocity coupling w/ bs 5 (f) data & velocity coupling w/ bs 100

Figure 1: The generated velocity distributions at $(x_t, t) = (-1, 0)$ for the dataset 1D $\mathcal{N} \to 2\mathcal{N}$ using HRF2, (a) without couplings, (b)-(c) with data coupling (batch sizes: 5 and 100), (d) with velocity coupling (batch size: 100), (e)-(f) with velocity coupling (batch size: 100) and data coupling (batch sizes: 5 and 100). Data coupling simplifies the velocity distribution (cf. (a)-(c)), while velocity coupling reduces the number of sampling steps.

$x_0$ and data points $x_1$ are no longer combined randomly into pairs $(x_0, x_1)$. We find that this controls multimodality of the distributions, making them easier to learn and improving overall model performance. Notably, coupling at one hierarchy level simplifies the distributions at all lower levels, thereby facilitating the matching process at the current level. Importantly, the complexity of the learned distribution can be controlled by adjusting the batch size in the mini-batch coupling process.

To illustrate the aforementioned distribution simplification, we provide an example with 1D data. In this example, the source distribution is a standard Gaussian, while the target distribution is a mixture of two Gaussians with means located at $-1$ and $1$. As shown in Figure 1(a-c), after applying data coupling (depth 1), the velocity distribution (depth 2) collapses into a single-mode Gaussian as the coupling batch size (bs) increases, effectively simplifying the velocity layer's distribution. The number given in the legend refers to the number of used velocity ODE integration steps.

From Figure 1(d), we observe that velocity coupling on its own does not simplify the velocity distribution. Instead, it simplifies the distribution at the next level (acceleration, not shown in the figure). Simplifying the acceleration distribution straightens the paths for velocity samples, reducing the number of integration steps needed to model the velocity distribution, as shown in the figure: 5 steps is almost as good as 100 steps. Figure 1(e,f) demonstrates that data coupling and velocity coupling are not mutually exclusive. They can be applied simultaneously to complement each other.

Formally, HRF was designed with independently sampled $x_0$ and $x_1$. In this paper, we first show that the underlying theory can be generalized to an arbitrary joint distribution over $x_0$ and $x_1$, i.e., $\gamma(x_0, x_1)$, which has the correct marginal distributions, i.e.,

$$\int \gamma(x_0, x_1) dx_1 = \rho_0(x_0) \text{ and } \int \gamma(x_0, x_1) dx_0 = \rho_1(x_1). \tag{8}$$

The following theorem characterizes the distribution of the velocity at a specific space-time location $(x_t, t)$ if an arbitrary joint distribution $\gamma$ is used instead of a product of two independent distributions.

**Theorem 3.1.** *The velocity distribution $\pi_1(v; x_t, t)$ at the space-time location $(x_t, t)$ induced by the linear interpolation in Equation (2) for $(x_0, x_1) \sim \gamma(x_0, x_1)$ is*

$$\pi_1(v; x_t, t) = \frac{\gamma(x_t - tv, x_t + (1-t)v)}{\rho_t(x_t)}, \tag{9}$$

| **Algorithm 1:** HRF2 with Data Coupling |
|---|
| **Input** : The source distributions $\rho_0$ and $\pi_0$, the dataset $\mathcal{D}$, and the batch size $B$. |
| 1 **while** *stopping conditions not satisfied* **do** |
| 2 $\quad$ Sample $\{x_0^{(i)}\}_{i=1}^B \sim \rho_0$, $\{x_1^{(i)}\}_{j=1}^B \sim \mathcal{D}$, and $\{v_0^{(i)}\}_{k=1}^B \sim \pi_0$; |
| 3 $\quad$ Sample $\{t^{(i)}\}_{i=1}^B \sim U[0,1]$ and $\{\tau^{(i)}\}_{i=1}^B \sim U[0,1]$; |
| 4 $\quad$ Use optimal transport to construct a set of coupled source and target pairs $\{(x_0^{(i)}, x_1^{(\sigma(i))})\}_{i=1}^B$; |
| 5 $\quad$ Compute loss following Equation (11); |
| 6 $\quad$ Perform gradient update on $\theta$; |
| 7 **end** |
| **Output** : $\theta$ |

| **Algorithm 2:** HRF2 with Velocity Coupling |
|---|
| **Input** : The source distributions $\rho_0$ and $\pi_0$, and the dataset $\mathcal{D}$ |
| 1 **while** *stopping conditions not satisfied* **do** |
| 2 $\quad$ Sample $x_0 \sim \rho_0, x_1 \sim \mathcal{D}$, and $v_0 \sim \pi_0$; |
| 3 $\quad$ Sample $t \sim U[0,1]$ and $\tau \sim U[0,1]$; |
| 4 $\quad$ Compute coupled $v_0$ and $v_1(x_t, t)$ using Algorithm 3; |
| 5 $\quad$ Compute loss using Equation (12); |
| 6 $\quad$ Perform gradient update on $\theta$; |
| 7 **end** |
| **Output** : $\theta$ |

*where*

$$\rho_t(x_t) = \int \gamma(x_t - tv, x_t + (1-t)v)dv, \qquad (10)$$

*and $\rho_t(x_t) \neq 0$. The distribution $\pi_1(v; x_t, t)$ is undefined if $\rho_t(x_t) = 0$.*

The proof of Theorem 3.1 is deferred to Appendix A. In Appendix B, we provide theoretical analysis on 1D Gaussian mixtures to illustrate how the mini-batch OT with data coupling is able to simplify the original multimodal velocity distributions. In addition, Appendix C shows the distribution of the acceleration under velocity couplings. Combining these two results, we show that data coupling and velocity coupling can gradually simplify the acceleration distribution.

Next, we detail how data coupling and velocity coupling can be achieved.

### 3.2 HRF2 WITH DATA COUPLING

To simplify the velocity distribution by reducing its multimodality, it is crucial to understand the cause of multimodality. During training, if source data $x_0$ and target data $x_1$ are sampled independently, the multimodality inherent in the data is preserved in the velocity distribution at $t = 0$. As mentioned in Section 3.1, Zhang et al. (2025) showed this. Breaking this independence is hence key to simplifying the velocity distribution. We find that couplings that restrict flexibility, e.g., mini-batch OT, provide an opportunity to do this. Intuitively, using mini-batch OT results in a coupling of source and target data that is no longer arbitrary, which inherently simplifies the velocity distribution.

Following Tong et al. (2024); Pooladian et al. (2023), we apply mini-batch OT on the data used for HRF2 training. Let $\{x_0^{(i)}\}_{i=1}^B \sim \rho_0$ and $\{x_1^{(i)}\}_{i=1}^B \sim \mathcal{D}$. The OT problem in Equation (7) can be solved exactly and efficiently on a small batch size using standard solvers, e.g., POT (Flamary et al., 2021). The resulting coupling from the algorithm gives us a permutation matrix that pairs $x_0^{(i)}$ with $x_1^{(\sigma(i))}$ for $i \in \{1, \ldots, B\}$. Instead of sampling $x_0$ and $x_1$ independently from $\rho_0$ and the dataset $\mathcal{D}$ during training, we jointly sample pairs $(x_0, x_1)$ from the joint distribution $\gamma(x_0, x_1)$ characterized by the mini-batch OT result. Using these samples, the training objective reads as follows:

$$\min_\theta \mathbb{E}_{(x_0,x_1)\sim\gamma, t\sim U[0,1], v_0\sim\pi_0, \tau\sim U[0,1]} \left[ \|(x_1 - x_0 - v_0) - a_\theta(x_t, t, v_\tau, \tau)\|_2^2 \right]. \qquad (11)$$

The optimization procedure is detailed in Algorithm 1.

### 3.3 HRF2 WITH VELOCITY COUPLING

Similar to data coupling, velocity coupling also aims to eliminate the independence between $v_0$ and $v_1(x_t, t)$. With mini-batch coupled velocity samples that are drawn from an underlying joint distribution $\kappa_{x_t,t}(v_0, v_1(x_t, t))$, the corresponding objective function is defined as follows:

$$\min_\theta \mathbb{E}_{x_0\sim\rho_0, x_1\sim\mathcal{D}, (v_0,v_1)\sim\kappa_{x_t,t}, t\sim U[0,1], \tau\sim U[0,1]} \left[ \|(v_1(x_t, t) - v_0) - a_\theta(x_t, t, v_\tau, \tau)\|_2^2 \right]. \qquad (12)$$

---

**Algorithm 3:** Velocity Coupling via Mini-Batch OT

**Input** : Location $(x_t, t)$, source distribution $\pi_0$, batch size $B$, and a pre-trained HRF2 model $a_\theta$.

1  Sample $\{v_0^{(i)}\}_{i=1}^B \sim \pi_0$ ;
2  Generate $v_1^{(i)}(x_t, t)$ from $v_0^{(i)}$ via numerically solving Equation (4) with a pre-trained $a_\theta$ for
   $i = 1, \ldots, B$;
3  Use OT to couple source and target points;

**Output** : Coupled samples $\{(v_0^{(i)}, v_1^{(\sigma(i))}(x_t, t))\}_{i=1}^B$.

---

Algorithm 2 summarizes the optimization procedure, for which we study the following coupling.

**Velocity Coupling via Batch OT.** Different from the data coupling, where the target data samples are readily available, obtaining velocity samples for a fixed $(x_t, t)$ requires simulation. Note that the target velocity samples $v_1(x_t, t) \sim \pi_1(v; x_t, t)$. To correctly couple $v_0$ and $v_1(x_t, t)$, it is essential to fix the space-time location $(x_t, t)$, ensuring that the samples $v_1(x_t, t)$ are drawn from the same velocity distribution. Thus, the first step of velocity coupling is to obtain a batch of $v_1(x_t, t)$ at a fixed $(x_t, t)$. We achieve this by using a pre-trained HRF2 model $a_\theta(x_t, t, v_\tau, \tau)$, which transport $v_0$ to $v_1(x_t, t)$ according to Equation (4). We then use the 2-Wasserstein optimal transport to sample coupled pairs $(v_0, v_1(x_t, t))$. This is detailed in Algorithm 3.

### 3.4 HRF2 with Hierarchical Data & Velocity Couplings

As shown before, data and velocity coupling complement each other. To use both couplings we need $(x_0, x_1) \sim \gamma$ and $(v_0, v_1(x_t, t)) \sim \kappa_{x_t, t}$. To achieve this, we apply a two-stage training. First, we use data coupling to train the model $a_\theta$ according to Algorithm 1. In the second stage, using this pre-trained model, we generate paired samples $(v_0, v_1(x_t, t))$ according to Algorithm 3. We then train our network using the following objective:

$$\min_\theta \mathbb{E}_{(x_0, x_1) \sim \gamma, (v_0, v_1) \sim \kappa_{x_t, t}, t \sim U[0,1], \tau \sim U[0,1]} \left[ \|(v_1(x_t, t) - v_0) - a_\theta(x_t, t, v_\tau, \tau)\|_2^2 \right]. \quad (13)$$

Importantly, note that the coupling of the velocities depends on the coupling of the data.

### 3.5 Marginal Preserving Property

The consistency of the velocity distribution with mini-batch velocity coupling directly follows prior works that use mini-batch coupling and reflow for data generation (Tong et al., 2024; Pooladian et al., 2023).

In addition, we can prove that the generation process according to Equation (6) with trained $a_\theta$ using mini-batch data coupling preserves the target data distribution and leads to correct marginals for all times $t \in [0, 1]$.

**Theorem 3.2.** *The time-differentiable stochastic process $\boldsymbol{Z} = \{Z_t : t \in [0, 1]\}$ generated by Equation (6) has the same marginal distribution as the time-differentiable stochastic process $\boldsymbol{X} = \{X_t : t \in [0, 1]\}$ generated by the linear interpolation in Equation (2) with the joint distribution $\gamma$ induced by mini-batch coupling.*

The proof of Theorem 3.2 is deferred to Section D.

## 4 Experiments

In this section, we explore how data coupling and velocity coupling influence the velocity distribution and assess whether simplifying it leads to performance improvements. For all experiments, we report the total neural function evaluations (NFEs), which represents the product of the number of integration steps across all HRF levels.

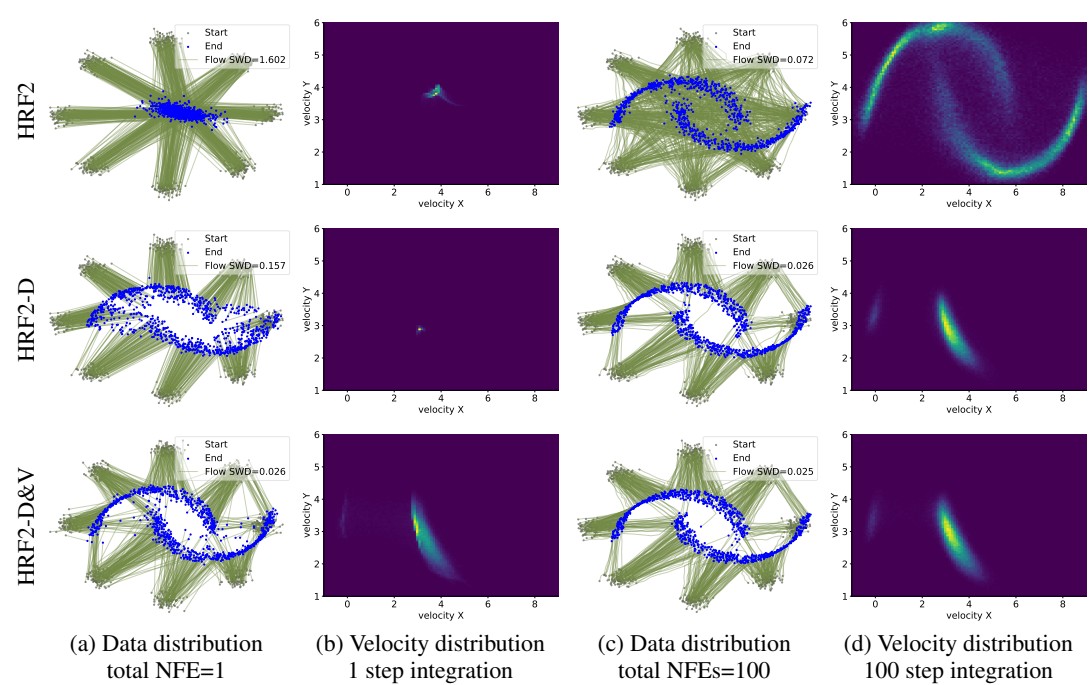

(a) Data distribution
total NFE=1

(b) Velocity distribution
1 step integration

(c) Data distribution
total NFEs=100

(d) Velocity distribution
100 step integration

Figure 2: Results on $8\mathcal{N} \to$ moon dataset. Three rows are HRF2, HRF2 with data coupling, HRF2 with data & velocity coupling. (a) and (c) are trajectories (green) of sample particles flowing from source distribution (grey) to target distribution (blue) with total NFEs 1 and 100. (b) and (d) are velocity distributions at the center of the bottom left Gaussian mode at $t = 0$. Data coupling simplifies the velocity distribution and velocity coupling reduces the required sampling steps.

## 4.1 SYNTHETIC DATA

We conduct experiments on four synthetic datasets used by Zhang et al. (2025) to ensure a fair comparison with HRF2. These datasets include two 1D cases ($\mathcal{N} \to 2\mathcal{N}$, $\mathcal{N} \to 5\mathcal{N}$), and two 2D cases ($\mathcal{N} \to 6\mathcal{N}$ and $8\mathcal{N} \to$ moon-shaped data). We use the Wasserstein and sliced 2-Wasserstein distances to evaluate 1D and 2D experiments, respectively. A complete description of the model architecture, parameter settings, and training details is provided in Section F.1.

Recall Figure 1: for the 1D $\mathcal{N} \to 2\mathcal{N}$ dataset we observed that data coupling simplifies the velocity distribution. Now, we extend this analysis to 2D datasets. We denote HRF2 with data coupling as HRF2-D and HRF2 with joint data and velocity coupling as HRF2-D&V. As shown in Figure 2, the 2D results corroborate the 1D findings. For the original HRF2, the velocity distribution at $t = 0$ is simply a shifted version of the data distribution. After applying data coupling, the velocity distribution at a given space-time location $(x_t, t)$ becomes more unimodal, effectively aligning with a portion of the target distribution. For example, in Figure 2(d), we observe that the velocity distribution primarily consists of the region of the target distribution closest to $x_t$.

In contrast, velocity coupling does not modify the velocity distribution itself but significantly reduces the number of required sampling steps. As shown in Figure 2(b), a single integration step already produces a reasonable velocity distribution with joint data and velocity couplings. The results in Figure 3 demonstrate that joint data and velocity couplings effectively enhance the model performance, particularly when NFEs are low. Additional results on synthetic data are provided in Section E.1.

## 4.2 IMAGE DATA

For high-dimensional image data, we conduct experiments on MNIST (LeCun et al., 1998), CIFAR-10 (Krizhevsky, 2009), and CelebA-HQ 256 (Karras et al., 2018), using Fréchet Inception Distance (FID) as the evaluation metric. For MNIST and CIFAR-10, we directly operate in the pixel space,

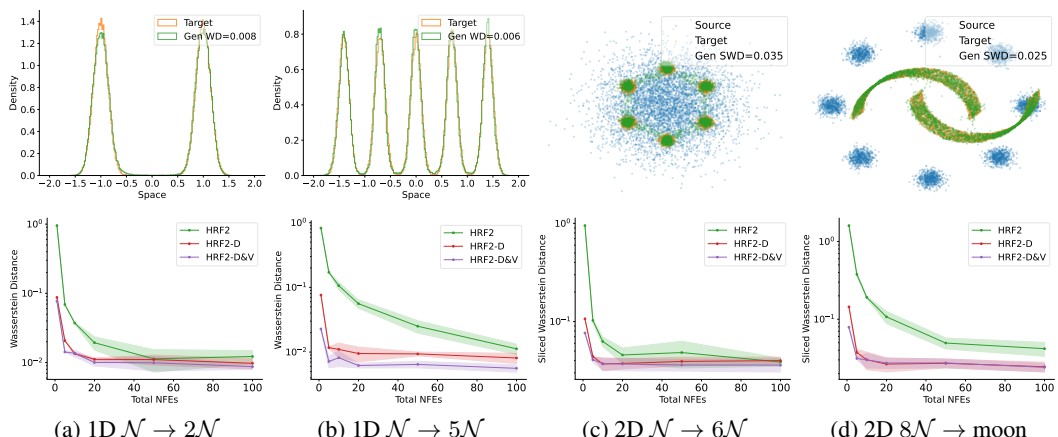

Figure 3: Results on synthetic datasets: (a) 1D $\mathcal{N} \to 2\mathcal{N}$ (b) 1D $\mathcal{N} \to 5\mathcal{N}$ (c) 2D $\mathcal{N} \to 6\mathcal{N}$ (d) 2D $8\mathcal{N} \to$ moon. Top row: HRF2-D&V generated data distributions. Bottom row: performance vs. total NFEs. We use Wasserstein and sliced 2-Wasserstein distances for 1D and 2D data, respectively.

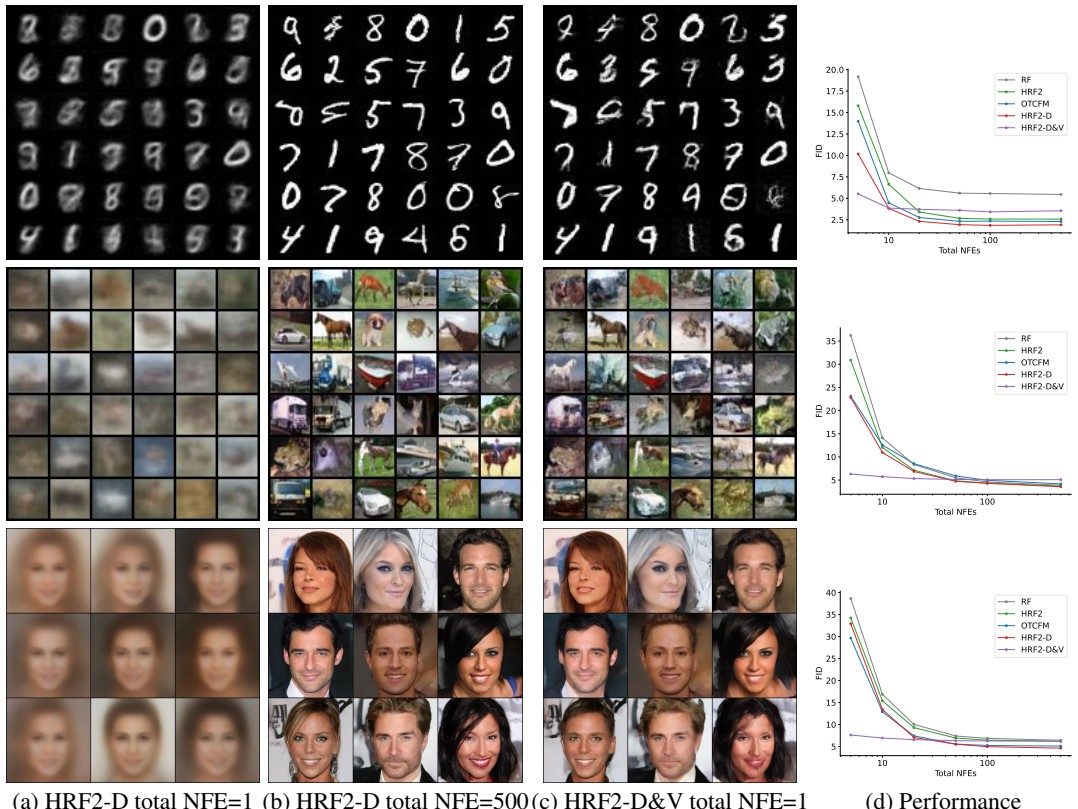

Figure 4: Results on MNIST, CIFAR-10 and CelebA-HQ 256 datasets. (a) HRF2-D with total NFE=1. (b) HRF2-D with total NFE=500. (c) HRF2-D&V with total NFE=1. Here we report the results with HRF2-D&V-OT. (d) FID scores with respect to total NFEs. With joint data coupling and velocity coupling, HRF2-D&V can generate reasonably good results with only 1 step.

with input dimensions of $1 \times 28 \times 28$ and $3 \times 32 \times 32$, respectively. For CelebA-HQ, we first encode the original $3 \times 256 \times 256$ images into a $4 \times 32 \times 32$ latent space using a pretrained VAE encoder, and conduct training and inference in the latent space. The experiments on CelebA-HQ demonstrate that the methods scale well on higher-dimensional data.

We compare our method to RF (Liu, 2022), HRF2 (Zhang et al., 2025) and OT-CFM (Tong et al., 2024) baselines. HRF2 is the base pre-trained model for our HRF2-D and HRF2-D&V. OT-CFM is essentially equivalent to HRF1-D. As shown in Figure 4(d), data coupling significantly improves performance across both low and high total NFEs. However, applying velocity coupling on top of data coupling only yields substantial improvements in the low-NFE regime. From Figure 4(a)-(c), we observe that data coupling alone enhances performance at low NFEs, but HRF2-D still struggles in extreme cases. Notably, incorporating velocity coupling enables the model to generate compelling results even under the extreme condition of total NFE = 1. More results are presented in Section E.2.

The details of the model architectures are deferred to Section F.2. Since the model architecture remains unchanged, HRF2-D and HRF2-D&V have the same memory usage and inference time as HRF2. For training, we apply data coupling and velocity coupling following Algorithms 1 and 2. To obtain accurate velocity pairs $(v_0, v_1)$ for velocity coupling, more integration steps are required here than for synthetic data. More details are provided in Section F.2.

## 5 RELATED WORK

**Flow Matching:** Concurrently, Liu et al. (2023a); Lipman et al. (2023); Albergo & Vanden-Eijnden (2023) presented learning of the ODE velocity that governs the generation of new data through a time-differentiable stochastic process defined by interpolating between samples from the source and data distributions. This provides flexibility by enabling precise connections between any two densities over finite time intervals. Liu et al. (2023a) concentrated on a linear interpolation, which provides straight paths connecting points from the source and the target distributions. Lipman et al. (2023) introduced the interpolation through the lens of conditional probability paths leading to a Gaussian. Albergo & Vanden-Eijnden (2023); Albergo et al. (2023) introduced stochastic interpolants with more general forms. They all learn the expected velocity field, which leads to curved sampling paths for data generation. Flow matching has been extended to handle discrete data (Gat et al., 2024; Stark et al., 2024) and manifold data (Chen & Lipman, 2024).

**Straightening Flows:** Liu et al. (2023a) proposed an iterative method called reflow, which connects points from the source and target distributions using a trained rectified flow model to smooth the transport path. They demonstrated that repeating this process results in an optimal transport map. However, in practice, errors in the learned velocity field can introduce bias. Other related studies address this by adjusting how noise and data are sampled during training, rather than using iterations. For instance, Pooladian et al. (2023); Tong et al. (2024) computed mini-batch optimal transport couplings between the source and data distributions to reduce transport costs and flow variance. Park et al. (2024) address the curved paths in flow matching by learning both initial velocity and acceleration, such that the sampling paths can cross. However, it requires a pre-trained diffusion model to acquire noise-data pairs. Cheng & Schwing (2025) study conditional data.

**Distribution of flow fields:** Zhang et al. (2025) capture the distribution of the random flow fields induced by the linear interpolation of source and target data. The sampling process is governed by coupled ODEs, which allows sampling paths to cross. Guo & Schwing (2025) model the flow field distribution using a variational autoencoder.

Building upon work by Zhang et al. (2025), we show that modeling the velocity distributions after the mini-batch coupling in data space improves the performance of HRF2 and OT-CFM. Hierarchically coupling the data and velocity leads to significantly improved results at low NFEs.

## 6 CONCLUSION

We study ways to control the complexity of the multimodal velocity distribution and their impact on capturing this distribution with hierarchical flow matching. We find hierarchical flow matching with mini-batch coupling in the data space consistently improves the generation quality compared to vanilla hierarchical rectified flow matching and vanilla flow matching with mini-batch optimal transport. Joint coupling in the data space and the velocity space leads to further improvements if few function evaluations are used. Code will be released for reproducibility of the results.

**Limitations and broader impacts:** Our proposed methods offer faster and more accurate data generation. It can help advance scientific modeling and simulations, contributing to advances in

areas like physics, healthcare, and drug discovery. For the limitations: the current velocity coupling approach requires simulated target velocity samples during training. Simulation-free velocity coupling is an interesting direction for future research.

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

## APPENDIX: HIERARCHICAL RECTIFIED FLOW MATCHING WITH MINI-BATCH COUPLINGS

This appendix is structured as follows: in Section A, we provide the proof of Theorem 3.1; in Section B, we show how mini-batch OT simplifies the 1D velocity distributions; in Section C, we assess acceleration distributions under couplings; in Section D we provide the proof of Theorem 3.2; in Section E we provide additional results for both synthetic and image data; in Section F we discuss implementation details.

## A    PROOF OF THEOREM 3.1

**Proof of Theorem 3.1:** For simplicity, we show the proof for 1D random variables $x_0$ and $x_1$ drawn from a joint distribution $\gamma(x_0, x_1)$. The joint distribution of $v(x_t, t)$ and $x_t$ is $\pi_1(v; x_t, t)\rho_t(x_t)$, since $\pi_1(v; x_t, t)$ corresponds to the conditional distribution of the velocity given location $x_t$. According to the linear interpolation in Equation (2), we have

$$\begin{bmatrix} v(x_t, t) \\ x_t \end{bmatrix} = \begin{bmatrix} 1 & -1 \\ t & (1-t) \end{bmatrix} \begin{bmatrix} x_1 \\ x_0 \end{bmatrix} = A \begin{bmatrix} x_1 \\ x_0 \end{bmatrix}, \tag{14}$$

where the matrix $A$ has determinant 1. Since $[v(x_t, t), x_t]^T$ is a linear transformation of $[x_1, x_0]^T$, we have the following expression for the joint distribution of $v(x_t, t)$ and $x_t$:

$$\pi_1(v; x_t, t)\rho_t(x_t) = \frac{1}{\det(A)} \gamma \left( A^{-1} \begin{bmatrix} v(x_t, t) \\ x_t \end{bmatrix} \right) = \gamma(x_t - tv, x_t + (1-t)v). \tag{15}$$

After rearranging, we get $\pi_1(v; x_t, t) = \frac{\gamma(x_t - tv, x_t + (1-t)v)}{\rho_t(x_t)}$. For the higher dimensional case, the relation in Equation (15) still holds. This completes the proof. ∎

## B    MINI-BATCH OT ANALYSIS

Here we use 1D distributions to illustrate how mini-batch OT in the data space simplifies the velocity distributions. We have the following result:

**Theorem B.1.** *Let $\rho_0$ be a standard 1D Gaussian distribution, and $\rho_1$ be a mixture of two well-separated Gaussians, i.e., $\rho_1 = \frac{1}{2}\mathcal{N}(-1, \sigma^2) + \frac{1}{2}\mathcal{N}(1, \sigma^2)$ and $\sigma \ll 1$. Let $B$ denote the batch size, with $B = 2k$ for some positive integer $k$. Let $\delta_1$ and $\delta_2$ be non-negative functions of $k$ such that $\delta_1 k \to 0$ and $\delta_2^2 k \to 0$ when $k \to \infty$. Then there exist positive constants $c_1, c_2$ with probability at least $1 - 2\exp(-c_1\delta_1^2 k) - 2\exp(-c_2\delta_2^2 k)$, such that the velocity distribution $\pi_1(v; x, t)$ is unimodal after mini-batch OT matching for $x \notin [l, u]$, where $l$ and $u$ depend on $k$, $\delta_1$, $\delta_2$, and $t$. The probability that $x \in [l, u]$ is $\int_l^u \rho_t(x)dx$. As $k \to \infty$, $l = u$ and the velocity distributions are unimodal almost surely.*

**Proof of Theorem B.1:**

Let $\{x_0^i\}_{i=1}^B$ and $\{x_1^i\}_{i=1}^B$ be a mini-batch of data points drawn independently from the source distribution $\rho_0$ and the target $\rho_1$ respectively. Without loss of generality, we assume that the target mixture distribution is well-separated such that for each batch the largest sample drawn from the left mode $\mathcal{N}(-1, \sigma^2)$ is smaller than the smallest sample drawn from the right mode $\mathcal{N}(1, \sigma^2)$ with high probability.

Let $\{x_0^{(i)}\}_{i=1}^B$ and $\{x_1^{(i)}\}_{i=1}^B$ be the respective ordered statistics. That is $x_0^{(1)} \le x_0^{(2)} \le \cdots \le x_0^{(B)}$ and $x_1^{(1)} \le x_1^{(2)} \le \cdots \le x_1^{(B)}$. Thus the optimal transport maps $x_0^{(i)}$ to $x_1^{(i)}$ for all $i \in [B]$. We define a jump as $J = \min\{i \in [B] : x_1^{(i)} \sim \mathcal{N}(-1, \sigma^2), x_1^{(i+1)} \sim \mathcal{N}(1, \sigma^2)\}$.

We first consider velocity $v$ at $t = 0$. Note that for each mini-batch, $v^{(b)}(x_0^{(i)}, 0) = x_1^{(i)} - x_0^{(i)}$ is always unimodal after an OT match, as it is deterministic. However, multimodality takes place if $\{v^{(b)}(x_0, 0)\}_{b \in [M]}$ points to different GMM modes in a neighborhood of $x_0$. In other words, multimodality of $\pi(v; x_0, 0)$ is in fact caused by randomness among different batches. For example

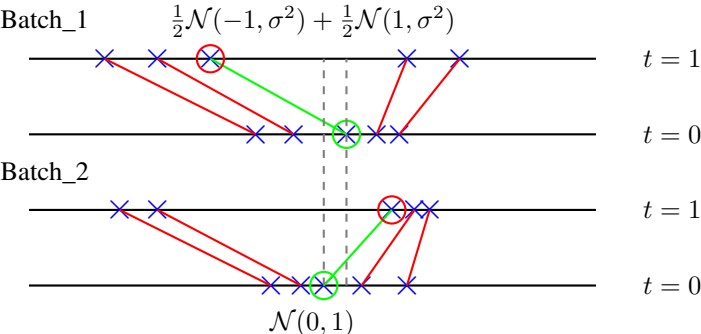

Figure 5: OT mapping of points between $t = 1$ and $t = 0$ for two mini-batches in red line. The green-circled samples are $x_0^{(J_1)}$ and $x_0^{(J_2+1)}$. The velocity distribution is $\pi(v; x, 0)$ is multimodal for $x \in \left[x_0^{(J_2+1)}, x_0^{(J_1)}\right]$.

in two different batches, whenever $x_0^{(J_1)} > x_0^{(J_2+1)}$, $\pi(v; x_0, 0)$ is multimodal in this region, see Figure 5 for an intuitive graphical illustration. Thus, we aim to control $x_0^{(J)}$ to restrict the region of $x_0$ that have multimodal velocity distributions. To control $x_0^{(J)}$, we start with providing a concentration bound on $J$ in the following lemma.

**Lemma B.2.** *For $\delta_1 > 0$ there exists $c_1 > 0$ such that with probability at least $1 - 2\exp(-c_1\delta_1^2 k)$,*
$$|J - k| \geq \delta_1 k. \tag{16}$$

*Proof.* By definition, $J$ is essentially the number of samples drawn from the first Gaussian mode, which follows a $\text{Binomial}(B, 1/2)$. Thus, $\mathbb{E}J = k$. By Chernoff's inequality (Vershynin, 2018), we have for $\delta_1 \in (0, 1]$,
$$\mathbb{P}(|X - \mu| \geq \delta_1\mu) \leq 2\exp(-c_1\mu\delta_1^2),$$
where $X$ is binomial and $\mu = \mathbb{E}X$. Thus, with probability at least $1 - 2\exp(-c_1\delta_1^2 k)$, we have $|J - k| \geq \delta_1 k$. □

The above lemma suggests that we lower bound $x_0^{((1-\delta_1)k)}$ and upper bound $x_0^{((1+\delta_1)k)}$ respectively, where we utilize the following lemma.

**Lemma B.3.** *Let $X_1, \ldots, X_N$ be i.i.d. random variables with CDF $F$. Let $X_{(1)}, \ldots, X_{(N)}$ be the ordered statistics. Then $\forall r \in [N]$,*
$$F(X_{(r)}) \sim \text{Beta}(r, N + 1 - r). \tag{17}$$

*Proof.* Let $F$ and $f$ be the CDF and PDF of $X$. We first show the CDF of the $r$-th ordered statistic $X_{(r)}$.

$$F_{(r)}(x) = \mathbb{P}(\text{at least } r \ X_i \leq x) = \sum_{j=r}^{N} \binom{N}{j} F(x)^j [1 - F(x)]^{N-j}.$$

Hence the PDF is
$$f_{(r)}(x) = \frac{d}{dx}F_{(r)}(x) = \frac{N!}{(r-1)!(N-r)!}F(x)^{r-1}[1 - F(x)]^{N-r}f(x).$$

Let $U = F(X_{(r)})$, so that $U \in [0, 1]$. Using change of variable, $u = F(x)$, the PDF $f_U$ is

$$f_U(u) = f_{(r)}(x)\left|\frac{dx}{du}\right|_{x=F^{-1}(u)}$$

$$\propto F(x)^{r-1}[1 - F(x)]^{N-r}f(x)\frac{1}{f(x)}$$

$$\propto u^{r-1}(1 - u)^{N-r},$$

which is a $\text{Beta}(r, N + 1 - r)$ distribution. $\qquad\square$

Lemma B.3 shows that $\mathbb{P}(X_{(r)} \geq a) = \mathbb{P}(F(X_{(r)}) \geq F(a))$, where the later tail probability could be bounded using concentration of the Beta distribution. We state it in the following lemma, which is a direct result of (Skorski, 2023).

**Lemma B.4.** *Let $F$ be the CDF of a continuous random variable $x$. For some $\delta_1 \in (0, 1)$ and $\delta_2 > 0$, let $l = F^{-1}(\mathbb{E}(F(x^{((1-\delta_1)k)})) - \delta_2)$ and $u = F^{-1}(\mathbb{E}(F(x^{((1+\delta_1)k)})) + \delta_2)$. Then there exist positive constants $C_1, C_2$ such that with probability at least $1 - \exp(-C_1\delta_2^2 k)$ and $1 - \exp(-C_2\delta_2^2 k)$: $x^{((1-\delta_1)k)} > l$ and $x^{((1+\delta_1)k)} < u$ respectively. Further, $\lim_{k\to\infty} l = \lim_{k\to\infty} u$ if $\delta_1, \delta_2$ are chosen such that $\delta_1 k \to 0$ and $\delta_2^2 k \to 0$ when $k \to \infty$.*

*Proof.* The proof is an application of Theorem 1 in (Skorski, 2023). Without loss of generality, we assume $(1 \pm \delta_1)k$ are integers. Now $U_1 = F(x_{((1-\delta_1))k}) \sim \text{Beta}((1 - \delta_1)k, (1 + \delta_1)k + 1)$. Thus

$$\mathbb{P}(U_1 < \mathbb{E}U_1 - \delta_2) \leq \exp(-\delta_2^2/2\theta),$$

where $\theta = \frac{(1-\delta_1)k[(1+\delta_1)k+1]}{(2k+1)^2(2k+2)} \simeq C_1/k$ for some absolute constant $C_1 > 0$. Denoting $l = F^{-1}(\mathbb{E}(F(x^{((1-\delta_1)k)})) - \delta_2)$, we have with probability at least $1 - \exp(-C_1\delta_2^2 k)$, $x^{((1-\delta_1)k)} > l$. A similar proof leads to the upper bound $u$.

If $\delta_1, \delta_2$ are chosen such that $\delta_1 k \to 0$ and $\delta_2^2 k \to 0$ when $k \to \infty$, then $\delta_2 \to 0$ and $(1 \pm \delta_1)k \sim k$. Thus $l = u = F^{-1}(\mathbb{E}(F(x^k))) = \mathbb{E}x$ in the limit. $\qquad\square$

Combining Lemma B.2 and Lemma B.4, the multimodal region of $\pi_1(v; x_0, 0)$ is bounded by $[l, u]$ with probability at least $1 - 2\exp(-c_1\delta_1^2 k) - 2\exp(-c_2\delta_2^2 k)$ for some absolute constant $c_1, c_2$. The probability that $x_0 \in [l, u]$ is $\int_l^u \rho_0(x)dx$. By further choosing $\delta_1, \delta_2$ such that $\delta_1 k \to 0$ and $\delta_2^2 k \to 0$ when $k \to \infty$, $l = u$ in the limit, eliminating all multimodality almost surely.

Since there is a bijection between $(x_0, x_1)$ and $(x_t, x_1)$ for all $t \in (0, 1)$, OT matching has the same effect on $(x_t, x_1)$ as on $(x_0, x_1)$. The joint distribution of $(x_t, x_1)$ is uniquely determined by $(x_0, x_1)$. Thus the proof on $(x_0, x_1)$ can be used for $(x_t, x_1)$, completing the proof of Theorem B.1.

A direct extension of Theorem B.1 is as follows.

**Corollary B.5.** *Let the target $\rho_1$ be a uniform mixture of $K$ well-separated Gaussians, i.e., $\rho_1 = \frac{1}{K}\sum_{i=1}^{K} \mathcal{N}(\mu_i, \sigma^2)$. Under the same settings as in Theorem B.1, let $\delta_1$ and $\delta_2$ be non-negative functions of $k$ such that $\delta_1 k \to 0$ and $\delta_2^2 k \to 0$ wehn $k \to \infty$. Then there exist positive constants $c_1, c_2$, with probability at least $1 - 2(K - 1)\exp(-c_1\delta_1^2 k) - 2(K - 1)\exp(-c_2\delta_2^2 k)$, such that after mini-batch OT matching the velocity distribution $\pi_1(v; x, t)$ is unimodal for $x \notin \bigcup_{i=1}^{K-1}[l_i, u_i]$, where $\{l_i\}_{i=1}^{K-1}$ and $\{u_i\}_{i=1}^{K-1}$ depend on $k$, $\delta_1$, $\delta_2$, and $t$. The probability that $x \in \bigcup_{i=1}^{K-1}[l_i, u_i]$ is $\sum_{i=1}^{K-1}\int_{l_i}^{u_i}\rho_t(x)dx$. As $k \to \infty$, $l_i = u_i$ for all $i \in [K - 1]$, and the velocity distributions are unimodal almost surely.*

*Proof.* In view of Theorem B.1, we prove the Corollary by induction. We first view $\mathcal{N}(\mu_1, \sigma^2)$ as one mode and the rest jointly as another. Applying Theorem B.1, we have with probability at least $1 - 2\exp(-c_1\delta_1^2 k) - 2\exp(-c_2\delta_2^2 k)$ the multimodal region of $v$ is bounded by $(l_1, u_1)$. We then consider the left two modes $\mathcal{N}(\mu_1, \sigma^2)$ and $\mathcal{N}(\mu_2, \sigma^2)$ jointly as one mode, and the rest jointly as another. Again applying Theorem B.1 we have with probability at least $1 - 2\exp(-c_1\delta_1^2 k) - 2\exp(-c_2\delta_2^2 k)$ the multimodal region of $v$ is bounded by $(l_2, u_2)$. Thus by induction,

$$\mathbb{P}(v \text{ is multimodal}) \leq \mathbb{P}\left(x_t \notin \bigcup_{i=1}^{K-1}(l_i, u_i)\right)$$

$$\leq \sum_{i=1}^{K-1} \mathbb{P}\left(x_t \notin (l_i, u_i)\right)$$

$$\leq (K - 1)[2\exp(-c_1\delta_1^2 k) + 2\exp(-c_2\delta_2^2 k)].$$

In the above derivation, the second inequality uses the union bound, and the third uses the result of Theorem B.1. Similar to the proof of Lemma B.4, $l_i = u_i$ when $k \to \infty$. $\qquad\square$

## C  DISTRIBUTION OF ACCELERATION $a$

In this section, we derive the acceleration distribution induced by linearly interpolating source and target data samples $x_0$, $x_1$ and by linearly interpolating source and target velocity samples $v_0$ and $v(x_t, t)$. Here, $(v_0, x_0, x_1)$ are drawn from an underlying joint distribution $\gamma$. In addition, we consider the acceleration distribution under the velocity coupling $\kappa_{x_t,t}$ at $(x_t, t)$.

**Theorem C.1.** *The acceleration distribution $p(a; x_t, t, v_\tau, \tau)$ is*

$$p(a; x_t, t, v_\tau, \tau) = \frac{\gamma\left(v_\tau - \tau a, x_t - t(v_\tau + (1-\tau)a), x_t + (1-t)(v_\tau + (1-\tau)a)\right)}{p_{t,\tau}(x_t, v_\tau)}, \quad (18)$$

*given a location $(x_t, t, v_\tau, \tau)$ induced by linearly interpolating data and velocity from $(v_0, x_0, x_1) \sim \gamma$ drawn from a joint distribution $\gamma$ that satisfies*

$$\int \gamma(v_0, x_0, x_1)dx_0 dx_1 = \pi_0(v_0), \quad \int \gamma(v_0, x_0, x_1)dv_0 dx_1 = \rho_0(x_0), \quad \int \gamma(v_0, x_0, x_1)dv_0 dx_0 = \rho_1(x_1).$$
$$(19)$$

*Here,*

$$p_{t,\tau}(x_t, v_\tau) = \int \gamma\left(v_\tau - \tau a, x_t - t(v_\tau + (1-\tau)a), x_t + (1-t)(v_\tau + (1-\tau)a)\right) da. \quad (20)$$

*The distribution $p(a; x_t, t, v_\tau, \tau)$ is undefined if $p_{t,\tau}(x_t, v_\tau) = 0$.*

*Proof.* For simplicity, we show the proof for 1D random variables $v_0$, $x_0$ and $x_1$ drawn from a joint distribution $\gamma(v_0, x_0, x_1)$. The joint distribution of $a_\tau$, $v_\tau$, and $x_t$ is $p(a; x_t, t, v_\tau, \tau)p_{t,\tau}(x_t, v_\tau)$, since $p(a; x_t, t, v_\tau, \tau)$ corresponds to the conditional distribution of the acceleration given locations $x_t$ and $v_\tau$. According to the linear interpolation in Equation (2), we have

$$\begin{bmatrix} v_\tau \\ a \\ x_t \end{bmatrix} = \begin{bmatrix} 1-\tau & \tau & 0 \\ -1 & 1 & 0 \\ 0 & 0 & 1 \end{bmatrix} \begin{bmatrix} v_0 \\ v(x_t, t) \\ x_t \end{bmatrix} = \begin{bmatrix} 1-\tau & \tau & 0 \\ -1 & 1 & 0 \\ 0 & 0 & 1 \end{bmatrix} \begin{bmatrix} 1 & 0 & 0 \\ 0 & -1 & 1 \\ 0 & 1-t & t \end{bmatrix} \begin{bmatrix} v_0 \\ x_0 \\ x_1 \end{bmatrix}$$

$$= \begin{bmatrix} 1-\tau & -\tau & \tau \\ -1 & 1 & 1 \\ 0 & 1-t & t \end{bmatrix} \begin{bmatrix} v_0 \\ x_0 \\ x_1 \end{bmatrix} = A \begin{bmatrix} v_0 \\ x_0 \\ x_1 \end{bmatrix}, \quad (21)$$

where the matrix $A$ has determinant 1. Since $[v_\tau, a, x_t]^T$ is a linear transformation of $[v_0, x_0, x_1]^T$, we have the following expression for the joint distribution of $a$ and $(x_t, v_\tau)$:

$$p(a; x_t, t, v_\tau, \tau)p_{t,\tau}(x_t, v_\tau) = \frac{1}{\det(A)} \gamma\left(A^{-1} \begin{bmatrix} v_\tau \\ a_\tau \\ x_t \end{bmatrix}\right)$$

$$= \gamma(v_\tau - \tau a, -tv_\tau - t(1-\tau)a + x_t, (1-t)v_\tau + (1-t)(1-\tau)a + x_t). \quad (22)$$

After rearranging we get $p(a; x_t, t, v_\tau, \tau) = \frac{\gamma(v_\tau - \tau a, -tv_\tau - t(1-\tau)a + x_t, (1-t)v_\tau + (1-t)(1-\tau)a + x_t)}{p_{t,\tau}(x_t, v_\tau)}$. For the higher dimensional case, the relation in Equation (22) still holds. This completes the proof. □

Theorem C.1 is stated for a general form of coupling among three random variables $V_0$, $X_0$, and $X_1$. In practice, we focus on hierarchically coupling the data and the velocity, for which we have the following corollary.

**Corollary C.2.** *The acceleration distribution $p(a; x_t, t, v_\tau, \tau)$ is*

$$p(a; x_t, t, v_\tau, \tau) = \frac{\kappa_{x_t,t}(v_\tau - \tau a, v_\tau + (1-\tau)a)}{\rho_\tau(v_\tau)}, \quad (23)$$

*given location $(x_t, t, v_\tau, \tau)$ induced by linearly interpolating between $v_0$ and $v_1$ from $(v_0, v_1) \sim \kappa_{x_t,t}$ drawn from a joint distribution $\kappa$ that satisfies*

$$\int \kappa_{x_t,t}(v_0, v_1)dv_1 = \pi_0(v_0) \quad \text{and} \quad \int \kappa_{x_t,t}(v_0, v_1)dv_0 = \pi_1(v_1; x_t, t). \quad (24)$$

*Here, $\rho_\tau(v_\tau) = \int \kappa_{x_t,t}(v_\tau - \tau a, v_\tau + (1-\tau)a)da$. The distribution $p(a; x_t, t, v_\tau, \tau)$ is undefined if $\rho_\tau(v_\tau) = 0$.*

*Proof.* The proof strategy is similar to the proof of Theorem 3.1, replacing $x_t$ with $v_\tau$, $v(x_t, t)$ with $a(x_t, t, v_\tau, \tau)$, and $\gamma$ with $\kappa_{x_t, t}$. $\qquad\square$

Combining the results in Appendix B and Corollary C.2, we can see that the data coupling results in simpler (less multimodal) target velocity distributions. In addition, with the velocity coupling $\kappa_{x_t, t}$, we further simplify the acceleration distributions.

# D    PROOF OF THEOREM 3.2

**Proof of Theorem 3.2:** We consider the characteristic function of $Z_{t+\Delta t} = Z_t + V\Delta t$ for $t \in [0, 1]$ and $\Delta t \in [0, 1 - t]$, assuming that $Z_t$ has the same distribution as $X_t$. If the characteristic functions of $Z_{t+\Delta t}$ and $X_{t+\Delta t}$ agree, then $Z_{t+\Delta t}$ and $X_{t+\Delta t}$ have the same distribution.

To show this, we evaluate the characteristic function of $Z_{t+\Delta t}$,

$$
\begin{aligned}
\mathbb{E}\left[e^{i\langle k, Z_{t+\Delta t}\rangle}\right] &= \mathbb{E}_{\rho_t, \pi_1}\left[e^{i\langle k, X_t + V\Delta t\rangle}\right] \\
&= \int\int e^{i\langle k, x_t + v\Delta t\rangle} \pi_1(v; x_t, t)\rho_t(x_t)dvdx_t \\
&\overset{a}{=} \int\int e^{i\langle k, x_t + v\Delta t\rangle} \frac{\gamma(x_t - vt, x_t + (1-t)v)}{\rho_t(x_t)} \rho_t(x_t)dvdx_t \\
&= \int\int e^{i\langle k, (x_t + v\Delta t)\rangle} \gamma(x_t - tv, x_t + (1-t)v)dvdx_t \\
&\overset{b}{=} \int\int e^{i\langle k, (1-t-\Delta t)x_0 + (t+\Delta t)x_1\rangle} \gamma(x_0, x_1)dx_0dx_1 \\
&= \mathbb{E}_{\rho_{t+\Delta t}}\left[e^{i\langle k, X_{t+\Delta t}\rangle}\right].
\end{aligned}
\tag{25}
$$

We use the notation $\langle \cdot, \cdot \rangle$ to denote the inner product. Equality $a$ is valid due to Theorem 3.2. Equality $b$ holds because $x_0 = x_t - tv$ and $x_1 = x_t + (1-t)v$ due to the linear interpolation. Therefore, we find that $Z_{t+\Delta t}$ and $X_{t+\Delta t}$ follow the same distribution. In addition, since $Z_0$ and $X_0$ follow the same distribution $\rho_0$, we can conclude that $Z_t$ and $X_t$ follow the same marginal distribution at $t$ for $t \in [0, 1]$. This completes the proof. $\qquad\blacksquare$

# E    ADDITIONAL EXPERIMENTAL RESULTS

## E.1    SYNTHETIC DATA RESULTS

We present more results on synthetic data: Figure 6 for 1D $\mathcal{N} \to 2\mathcal{N}$ data, Figure 7 for 1D $\mathcal{N} \to 5\mathcal{N}$ data, and Figure 8 for 2D $\mathcal{N} \to 6\mathcal{N}$ data. Across all these experiments, we consistently observe that data coupling simplifies the velocity distribution, while velocity coupling significantly reduces the required sampling steps. This corroborates the findings discussed in the main paper.

## E.2    IMAGE DATA RESULTS

We present more results on image data: Table 1 for MNIST, Table 2 for CIFAR-10, and Table 3 and Figure 9 for CelebA-HQ 256. Again, we consistently observe that data coupling enhances sampling quality for both low and high total NFEs, but collapses when total NFE is reduced to 1, while velocity coupling produces high-quality samples even under this extreme case.

### E.2.1    PERFORMANCE GAINS ACROSS DATASETS OF INCREASING COMPLEXITY

We evaluate performance gains across datasets of increasing complexity (MNIST $\to$ CIFAR-10 $\to$ CelebA-HQ). As shown in Table 4, HRF2-D consistently improves generation quality, with no clear diminishing trend as the evaluation resolution increases. Moreover, as shown in Table 5, HRF2-D&V significantly boosts performance at fixed low NFE, achieving up to 74.3% improvement on CelebA-HQ. These results demonstrate that our method scales well and remains effective on high-dimensional datasets.

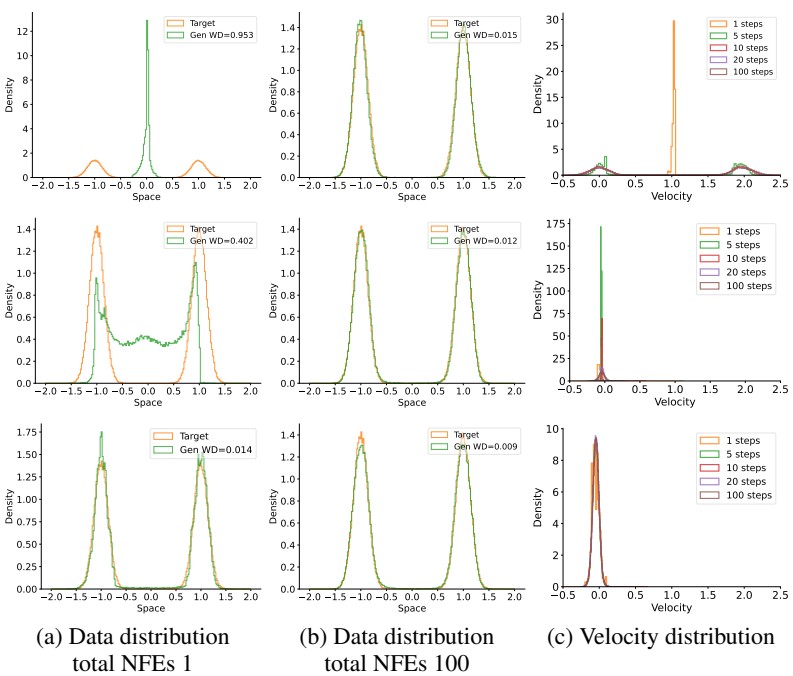

(a) Data distribution
total NFEs 1

(b) Data distribution
total NFEs 100

(c) Velocity distribution

Figure 6: Results on 1D $\mathcal{N} \rightarrow 2\mathcal{N}$ data. The three rows correspond to HRF2, HRF2 with data coupling, HRF2 with data & velocity coupling. (a) and (b) are generated data distribution with total NFEs 1 and 100. (c) is velocity distribution at $(x_t, t) = (-1, 0)$.

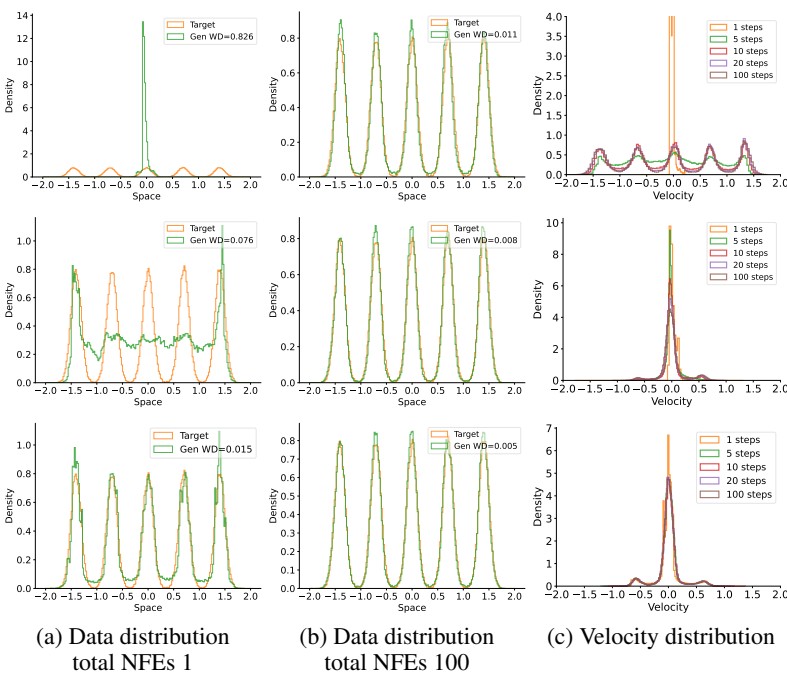

(a) Data distribution
total NFEs 1

(b) Data distribution
total NFEs 100

(c) Velocity distribution

Figure 7: Results on 1D $\mathcal{N} \rightarrow 5\mathcal{N}$ data. The three rows correspond to HRF2, HRF2 with data coupling, HRF2 with data & velocity coupling. (a) and (b) are generated data distribution with total NFEs 1 and 100. (c) is velocity distribution at $(x_t, t) = (0, 0)$.

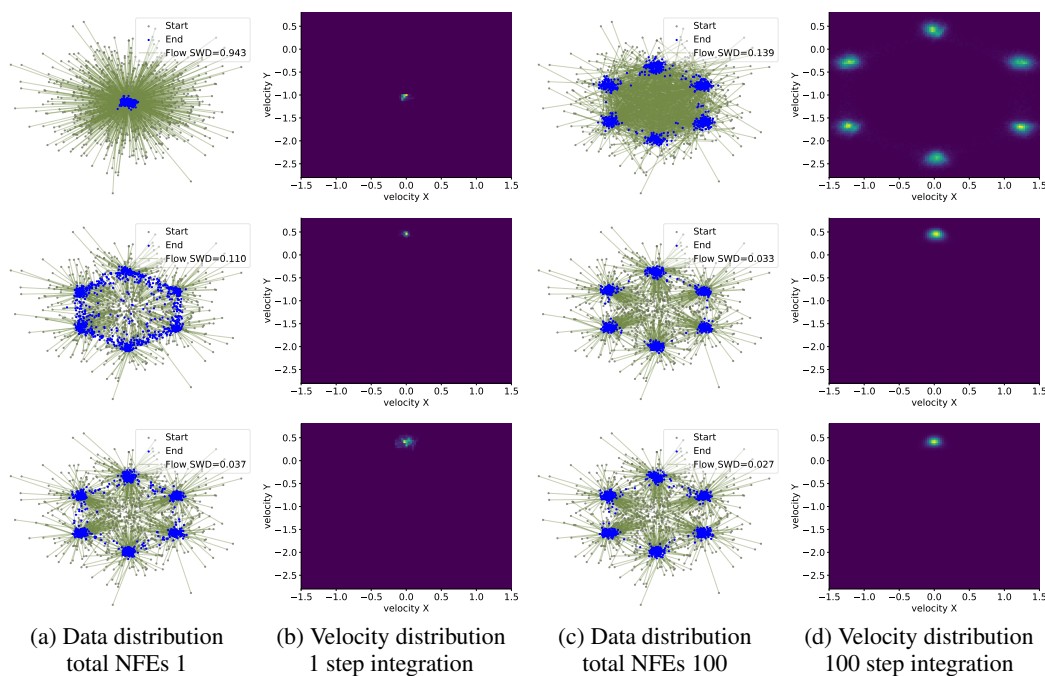

(a) Data distribution total NFEs 1

(b) Velocity distribution 1 step integration

(c) Data distribution total NFEs 100

(d) Velocity distribution 100 step integration

Figure 8: Results on 2D $\mathcal{N} \to 6\mathcal{N}$ data. The three rows correspond to HRF2, HRF2 with data coupling, HRF2 with data & velocity coupling. (a) and (c) are trajectories (green) of sample particles flowing from source distribution (grey) to target distribution (blue) with total NFEs 1 and 100. (b) and (d) are velocity distributions at $(0, 1)$ at $t = 0$.

Table 1: FID performance on MNIST under different total NFE settings. **Bold** for the best.

| Total NFEs | RF (1.08M) | OT-CFM (1.08M) | HRF2 (1.07M) | HRF2-D (1.07M) | HRF2-D&V (1.07M) |
|---|---|---|---|---|---|
| 5 | $19.187 \pm 0.188$ | $13.977 \pm 0.166$ | $15.798 \pm 0.151$ | $10.167 \pm 0.136$ | $\mathbf{5.519 \pm 0.112}$ |
| 10 | $7.974 \pm 0.119$ | $4.477 \pm 0.099$ | $6.644 \pm 0.076$ | $\mathbf{3.823 \pm 0.038}$ | $3.861 \pm 0.089$ |
| 20 | $6.151 \pm 0.090$ | $2.763 \pm 0.036$ | $3.408 \pm 0.076$ | $\mathbf{2.318 \pm 0.053}$ | $3.720 \pm 0.045$ |
| 50 | $5.605 \pm 0.057$ | $2.321 \pm 0.038$ | $2.664 \pm 0.058$ | $\mathbf{1.929 \pm 0.031}$ | $3.604 \pm 0.016$ |
| 100 | $5.563 \pm 0.049$ | $2.346 \pm 0.023$ | $2.588 \pm 0.075$ | $\mathbf{1.847 \pm 0.011}$ | $3.423 \pm 0.003$ |
| 500 | $5.453 \pm 0.047$ | $2.296 \pm 0.007$ | $2.574 \pm 0.121$ | $\mathbf{1.913 \pm 0.043}$ | $3.546 \pm 0.107$ |

### E.2.2 SUB-OPTIMAL CHECKPOINTS

Since HRF2-D is used to generate the training set for HRF2-D&V, one should expect that the quality of HRF2-D impacts the performance of HRF2-D&V. Our empirical results show that HRF2-D&V is relatively robust to the specific checkpoint, as long as the result quality is reasonable. In Table 6, we report the FID of the generated CIFAR-10 images at different training stages. It shows that using a suboptimal HRF2-D checkpoint yields similar performance to using the best checkpoint. This suggests that the second-stage training is robust to such variations.

### E.2.3 VELOCITY COUPLING WITH REFLOW

In our velocity coupling setting, if we directly use $(v_0, v_1)$ pairs generated from HRF2-D, it will be similar to the reflow process proposed by Liu (2022). We test and show in Table 7 that for CIFAR-10, using reflow for velocity coupling is slightly worse compared to velocity coupling with batch OT. Similar trends were observed on MNIST and CelebA-HQ data.

Table 2: FID performance on CIFAR-10 under different total NFE settings. **Bold** for the best.

| Total NFEs | RF (35.75M) | OT-CFM (35.75M) | HRF2 (44.81M) | HRF2-D (44.81M) | HRF2-D&V (44.81M) |
|---|---|---|---|---|---|
| 5 | 36.209 ± 0.142 | 23.111 ± 0.010 | 30.884 ± 0.104 | 22.817 ± 0.072 | **6.315 ± 0.057** |
| 10 | 14.113 ± 0.092 | 12.564 ± 0.016 | 12.065 ± 0.024 | 10.969 ± 0.025 | **5.739 ± 0.017** |
| 20 | 8.355 ± 0.065 | 8.553 ± 0.002 | 7.129 ± 0.027 | 6.860 ± 0.022 | **5.332 ± 0.009** |
| 50 | 5.514 ± 0.034 | 5.911 ± 0.005 | 4.847 ± 0.028 | **4.739 ± 0.006** | 5.142 ± 0.024 |
| 100 | 4.588 ± 0.013 | 4.952 ± 0.012 | 4.334 ± 0.054 | **4.301 ± 0.022** | 5.078 ± 0.044 |
| 500 | 3.887 ± 0.035 | 4.184 ± 0.086 | 3.706 ± 0.043 | **3.578 ± 0.028** | 5.095 ± 0.032 |

Table 3: FID performance on CelebA-HQ 256 under different total NFE settings. **Bold** for the best.

| Total NFEs | RF (457.06M) | OT-CFM (457.06M) | HRF2 (616.20M) | HRF2-D (616.20M) | HRF2-D&V (616.20M) |
|---|---|---|---|---|---|
| 5 | 38.641 ± 0.126 | 29.646 ± 0.093 | 34.246 ± 0.107 | 32.918 ± 0.085 | **7.612 ± 0.015** |
| 10 | 16.876 ± 0.088 | 12.879 ± 0.083 | 15.391 ± 0.074 | 13.424 ± 0.022 | **6.931 ± 0.038** |
| 20 | 10.027 ± 0.060 | 7.426 ± 0.042 | 9.291 ± 0.042 | 7.048 ± 0.033 | **6.560 ± 0.039** |
| 50 | 7.395 ± 0.021 | 5.545 ± 0.023 | 6.927 ± 0.041 | **5.529 ± 0.021** | 6.330 ± 0.022 |
| 100 | 6.850 ± 0.064 | 5.236 ± 0.034 | 6.450 ± 0.062 | **4.961 ± 0.019** | 6.248 ± 0.023 |
| 500 | 6.418 ± 0.026 | 5.094 ± 0.019 | 6.188 ± 0.056 | **4.624 ± 0.029** | 6.225 ± 0.015 |

# F    IMPLEMENTATION DETAILS

## F.1    SYNTHETIC DATA

For synthetic data experiments, we employ a neural network architecture with two distinct stages. The first stage separately encodes spatial and temporal inputs with linear layers and Sinusoidal Positional Embeddings. The second stage concatenates the processed features and refines them through multiple linear layers to produce the final output. The model consists of 304,513 parameters, totaling 0.30M in size.

For data coupling, we train the model from scratch following Algorithm 1 strictly. A key consideration is the choice of batch sizes, as two different batch sizes are involved – one for batch OT and another for training. In 1D and 2D experiments, a large batch size is necessary for stable training, but using an excessively large batch size for batch OT is computationally inefficient. To address this, we set the batch size for batch OT to 100 while using a batch size of 1,000 for gradient computation. This means that in each training iteration, we perform batch OT on 100 data points 10 times to accumulate a full batch for gradient updates.

For velocity coupling, we use the HRF2-D model from the previous step as the base model to generate $(v_0, v_1)$ pairs at a fixed space-time location $(x_t, t)$, following Algorithm 2. During training, we observed that the performance depends on the quality of the base model. To mitigate this, we save multiple checkpoints of HRF2-D and select the best-performing checkpoint via a validation dataset as the base model for velocity coupling.

Computational requirements during training are shown in Table 8. In the low-dimensional setting, batch OT becomes more time-consuming than the training itself. As a result, HRF2-D trains significantly slower than HRF2. In contrast, HRF2-D&V uses precomputed velocity pairs and therefore does not require batch OT during training. Moreover, it operates with a smaller batch size (1000) than HRF2 (5000), resulting in lower memory usage and faster training.

During the evaluation, we select the best checkpoint from a validation set for all models (HRF2, HRF2-D, HRF2-D&V). For each seed, we conduct the experiment three times, yielding three best models per seed. Each model is then evaluated three times, resulting in nine experimental results per seed. Finally, we report the mean and standard deviation over three different seeds, totaling 27 experimental results.

## F.2    IMAGE DATA

We adapt and modify the model architectures from Zhang et al. (2025) for MNIST and CIFAR-10 data and the model architecture from Dao et al. (2023) for the CelebA-HQ data.

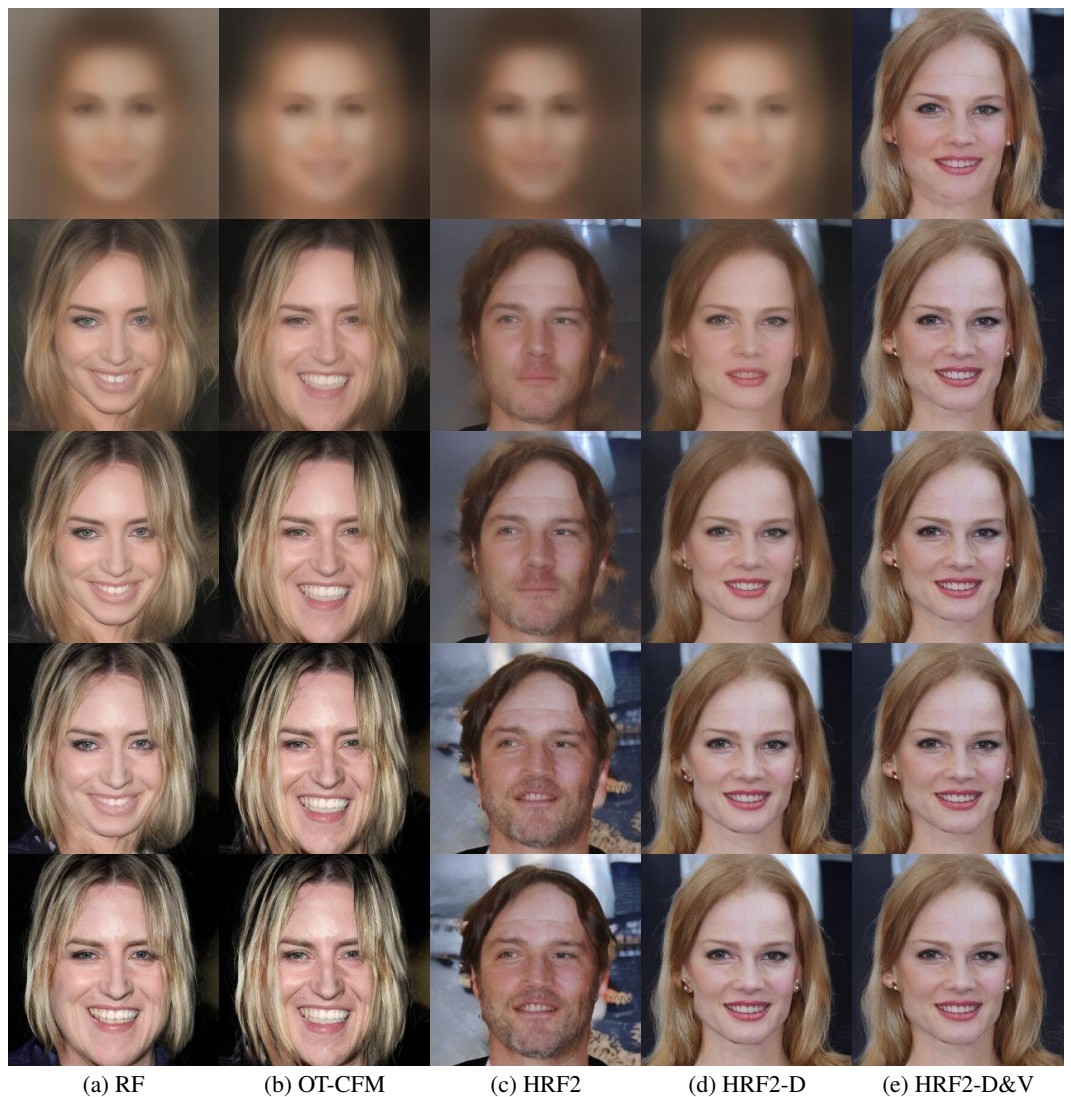

|          |           |          |           |            |
|----------|-----------|----------|-----------|------------|
| (a) RF   | (b) OT-CFM| (c) HRF2 | (d) HRF2-D| (e) HRF2-D&V|

Figure 9: Examples of the generated images for CelebA-HQ starting from the same noise for all models. The 5 rows from top to bottom correspond to total NFEs = 1, 5, 10, 50, 500.

Table 4: Best FID comparison of baseline and HRF2-D on different datasets. HRF2-D achieves consistent improvements over the baseline.

| Dataset    | Image Size                  | Baseline (best) | HRF2-D (best) | HRF2-D Improv. |
|------------|-----------------------------|-----------------|---------------|----------------|
| MNIST      | $1{\times}28{\times}28$     | 2.296           | 1.847         | 19.6%          |
| CIFAR-10   | $3{\times}32{\times}32$     | 3.706           | 3.578         | 3.5%           |
| CelebA-HQ  | $3{\times}256{\times}256$   | 5.094           | 4.624         | 9.2%           |

**MNIST.** For MNIST, we use the standard UNet. The ResNet blocks in the UNet function similarly to the model used for synthetic data. They process spatial and temporal inputs separately using convolutional and linear layers, respectively. The processed features are then concatenated and passed through a series of linear layers to capture space-time dependencies. The model consists of 1.07M parameters.

**CIFAR-10.** For CIFAR-10, the model consists of two UNets: a large UNet for processing $v_\tau$ and $\tau$, and a smaller UNet (one-fourth the size) for processing $x_t$ and $t$. The outputs of each ResNet block

Table 5: FID comparison of baseline and HRF2-D&V at NFE=5. Our HRF2-D&V model significantly outperforms the best baseline with large relative improvements.

| Dataset | Image Size | Best Baseline@NFE=5 | HRF2-D&V@NFE=5 | HRF2-D&V Improv. |
|---------|-----------|---------------------|-----------------|-------------------|
| MNIST | $1 \times 28 \times 28$ | 13.977 | 5.519 | 60.5% |
| CIFAR-10 | $3 \times 32 \times 32$ | 23.111 | 6.315 | 72.7% |
| CelebA-HQ | $3 \times 256 \times 256$ | 29.646 | 7.612 | 74.3% |

Table 6: Evaluation of HRF2-D&V model using HRF2-D at different training stages (steps). Reported values are the corresponding metric scores with NFE set to 100 and 5, respectively.

| Training Stage (steps) | HRF2-D NFE=100 | HRF2-D&V NFE=5 |
|------------------------|-----------------|------------------|
| Very early checkpoint (100k) | 6.935 | 9.326 |
| Later checkpoint (300k) | 4.672 | 6.833 |
| Latest checkpoint (400k) | 4.301 | 6.315 |

Table 7: Performance comparison with and without OT for velocity coupling across different NFE values. Our OT-based method consistently improves results over the velocity coupling with reflow.

| NFE | 5 | 10 | 20 | 50 | 100 | 500 |
|-----|---|----|----|----|-----|-----|
| w/o OT (reflow) | $6.632 \pm 0.074$ | $5.929 \pm 0.057$ | $5.669 \pm 0.036$ | $5.398 \pm 0.007$ | $5.273 \pm 0.017$ | $5.261 \pm 0.027$ |
| w/ OT (ours) | $6.315 \pm 0.057$ | $5.739 \pm 0.017$ | $5.332 \pm 0.009$ | $5.142 \pm 0.024$ | $5.078 \pm 0.044$ | $5.095 \pm 0.032$ |

in the smaller UNet are input to the corresponding ResNet blocks in the larger UNet, facilitating information exchange between different scales. The model consists of 44.81M parameters.

**CelebA-HQ.** For CelebA-HQ, we first encode images into a latent space using the pretrained VAE encoder from Stable Diffusion (Rombach et al., 2022). We then use DiT (Peebles & Xie, 2023) as the backbone to process $v_\tau$ in this latent space. To condition the velocity prediction on $x_t$, we inject $x_t$ into each DiT block via cross-attention layers, while keeping the main DiT architecture unchanged. The time embedding is also modified by replacing $\text{embedding}(t)$ with $\text{embedding}(t) + \text{embedding}(\tau)$ to incorporate time information in both time axes.

For training RF and OT-CFM on MNIST and CIFAR-10, we follow the procedures and hyperparameter settings from Tong et al. (2024) and Lipman et al. (2023). For HRF2 on the same datasets, we adopt the training setup from Zhang et al. (2025). For all models on CelebA-HQ, we follow the procedures and hyperparameters from Dao et al. (2023).

For data coupling, we train the model from scratch following Algorithm 1. Both the batch OT and training batch sizes are set to 128 for MNIST and CIFAR-10 and 256 for CelebA-HQ.

For velocity coupling, we start from the HRF2-D model obtained in the previous step. Following the synthetic data experiments, we select the best-performing model on the validation dataset to ensure training quality. The training speed for velocity coupling is primarily limited by the velocity sample generation. Therefore, we generate velocity pairs before training and perform the training offline.

We train the UNet for MNIST and CIFAR-10 on 1 NVIDIA L40S GPU and the DiT for CelebA-HQ on 8 NVIDIA L40S GPUs. Computational requirements, including training time and memory usage, are shown in Tables 9 to 11.

For each model, we conduct five evaluation runs, and report the means and standard deviations.

We use the `emd` function from the Python Optimal Transport (`pot`) library to compute exact OT. While the theoretical worst-case complexity is $O(n^3)$, we empirically observe much lower runtime scaling. As shown in Table 12, on CIFAR-10 data, the OT time grows sub-quadratically over batch sizes from 32 to 256, and remains negligible compared to a single training step (<2%).

Table 8: Computational requirements during training on synthetic datasets.

| Training | 1D data | | | 2D data | | |
|---|---|---|---|---|---|---|
| | HRF2 (0.30M) | HRF2-D (0.30M) | HRF2-D&V (0.30M) | HRF2 (0.32M) | HRF2-D (0.32M) | HRF2-D&V (0.32M) |
| Time (s/iter) | 0.0028 | 0.0581 | 0.0025 | 0.0029 | 0.0588 | 0.0027 |
| Memory (MB) | 658 | 658 | 566 | 660 | 660 | 568 |
| Param. Counts | 304,513 | 304,513 | 304,513 | 321,154 | 321,154 | 321,154 |

Table 9: Computational cost and model size for different methods on MNIST.

| MNIST | RF (1.08M) | OT-CFM (1.08M) | HRF2 (1.07M) | HRF2-D (1.07M) | HRF2-D&V (1.07M) |
|---|---|---|---|---|---|
| Time (s/iter) | 0.045 | 0.046 | 0.046 | 0.046 | 0.046 |
| Memory (MB) | 2546 | 2546 | 2546 | 2546 | 2546 |
| Param. Counts | 1,075,361 | 1,075,361 | 1,065,698 | 1,065,698 | 1,065,698 |

Table 10: Computational cost and model size for different methods on CIFAR-10.

| CIFAR-10 | RF (35.75M) | OT-CFM (35.75M) | HRF2 (44.81M) | HRF2-D (44.81M) | HRF2-D&V (44.81M) |
|---|---|---|---|---|---|
| Time (s/iter) | 0.166 | 0.169 | 0.196 | 0.202 | 0.200 |
| Memory (MB) | 7480 | 7480 | 9220 | 9220 | 9220 |
| Param. Counts | 35,746,307 | 35,746,307 | 44,807,843 | 44,807,843 | 44,807,843 |

# G INTEGRATING SHORTCUT MODELS INTO HIERARCHICAL RECTIFIED FLOW

Our data coupling and velocity coupling formulation provides a general framework that can be combined with any flow matching (FM) model. This is because the inner hierarchy of our coupled ODE in Equation (6) is a standard flow matching process, so any alternative parameterization of the acceleration field can be plugged in without modifying the hierarchical structure.

In this section, we use the ShortCut model (Frans et al., 2025) as an example to illustrate how distillation and one/few step FM algorithms can be incorporated into our setting. The ShortCut model introduces a desired step size $d$. The step size allows the model to anticipate future curvature and jump to the correct next point rather than drifting off the true trajectory. The one step update becomes

$$x_{t+d} = x_t + s_\theta(x_t, t, d)\, d,$$

where the model $s_\theta(x_t, t, d)$ learns shortcuts for all combinations of $x_t$, $t$, and $d$.

The training objective contains two terms: a standard flow matching loss and a self-consistency loss:

$$\mathbb{E}_{x_0 \sim \mathcal{N}(0,I),\, x_1 \sim \mathcal{D},\, t \sim U[0,1]\, d \sim p(d)} \left[ \|s_\theta(x_t, t, 0) - (x_1 - x_0)\|_2^2 + \|s_\theta(x_t, t, 2d) - s_{\text{target}}\|_2^2 \right],$$

$$\text{where} \quad s_{\text{target}} = s_\theta(x_t, t, d)/2 + s_\theta(x_{t+d}, t+d, d)/2.$$

To integrate this formulation into our hierarchical rectified flow, we replace the inner flow matching update of Equation (6) with the shortcut parameterization. The one step update becomes

$$v_{\tau+d} = v_\tau + s_\theta(x_t, t, v_\tau, \tau, d)\, d.$$

The corresponding objective is then

$$\mathbb{E}_{x_0 \sim \rho_0,\, x_1 \sim \mathcal{D},\, t \sim U[0,1],\, v_0 \sim \pi_0,\, \tau \sim U[0,1],\, d \sim p(d)}$$

$$\left[ \|s_\theta(x_t, t, v_\tau, \tau, 0) - (x_1 - x_0 - v_0)\|_2^2 + \|s_\theta(x_t, t, v_\tau, \tau, 2d) - s_{\text{target}}\|_2^2 \right],$$

$$\text{where} \quad s_{\text{target}} = s_\theta(x_t, t, v_\tau, \tau, d)/2 + s_\theta(x_t, t, v_{\tau+d}, \tau+d, d)/2.$$

Following the ShortCut model, the step size $d$ is drawn uniformly from the set $\{1/128, 1/64, \cdots, 1/2, 1\}$.

We evaluate this integration on the CIFAR-10 dataset. As shown in Table 13, Shortcut alone gives relatively high FID at one step, and combining the two consistently improves upon ShortCut. This confirms that the data and velocity coupling and shortcut-based distillation can be jointly used within our framework.

Table 11: Computational cost and model size for different methods on CelebA-HQ.

| CelebA-HQ 256 | RF (457.06M) | OT-CFM (457.06M) | HRF2 (616.20M) | HRF2-D (616.20M) | HRF2-D&V (616.20M) |
|---|---|---|---|---|---|
| Time (s/iter) | 0.418 | 0.420 | 0.688 | 0.688 | 0.688 |
| Memory per GPU (GB) | 26.65 | 26.65 | 39.89 | 39.89 | 36.79 |
| Param. Counts | 457,062,416 | 457,062,416 | 616,197,186 | 616,197,186 | 616,197,186 |

Table 12: Comparison of OT computation time and training step time across different batch sizes. OT time remains negligible compared to the cost of one training step.

| Batch Size | 32 | 64 | 128 | 256 |
|---|---|---|---|---|
| OT time (s) | 0.00078 | 0.00117 | 0.00215 | 0.00670 |
| One train step time (s) | 0.07 | 0.09 | 0.18 | 0.37 |

## H  ADAPTIVE SOLVERS

Our sampler consists of two nested integrations, and the inner integration follows a standard flow matching update. Since this inner step is independent of the hierarchical coupling structure, it can be replaced by any higher-order or adaptive ODE solver without modifying the formulation. This makes our framework compatible with existing adaptive solvers such as dopri5.

To illustrate this compatibility, we compare fixed step Euler sampling and adaptive dopri5 sampling on CIFAR-10 data (see Table 14). The results confirm that the hierarchical formulation does not restrict the choice of numerical solver and that adaptive solvers like dopri5 can be applied directly to the inner update. When an adaptive solver is used, HRF2-D continues to achieve the lowest FID among the compared methods. This shows that the benefits of hierarchical coupling are preserved regardless of the numerical solver.

## I  LLM USAGE

While preparing this work, we used a large language model (LLM) to assist with language editing. The LLM's contributions were limited to improving the clarity of the text. The core research, experimental design, and all scientific claims remain our original work.

Table 13: FID comparison on CIFAR-10 for the Shortcut model, HRF2-D&V, and the combined HRF2-D&V with Shortcut. The results show that Shortcut distillation and hierarchical coupling are complementary. Shortcut alone gives relatively high FID at one step, and combining the two consistently improves upon Shortcut.

| NFE | ShortCut | HRF2-D&V | HRF2-D&V + ShortCut |
|-----|----------|----------|---------------------|
| 1 | 41.60 | 15.17 | 16.95 |
| 4 | 15.78 | 6.78 | 11.35 |
| 8 | 12.68 | 6.06 | 10.45 |

Table 14: FID performance comparison between fixed step Euler sampling and adaptive dopri5 sampling on CIFAR-10 under different total NFE settings. **Bold** for the best.

| Total NFEs | RF (35.75M) | OT-CFM (35.75M) | HRF2 (44.81M) | HRF2-D (44.81M) | HRF2-D&V (44.81M) |
|------------|-------------|-----------------|---------------|-----------------|-------------------|
| 100 | $4.588 \pm 0.013$ | $4.952 \pm 0.012$ | $4.334 \pm 0.054$ | $\mathbf{4.301 \pm 0.022}$ | $5.078 \pm 0.044$ |
| 500 | $3.887 \pm 0.035$ | $4.184 \pm 0.086$ | $3.706 \pm 0.043$ | $\mathbf{3.578 \pm 0.028}$ | $5.095 \pm 0.032$ |
| Adaptive | $3.688 \pm 0.077$ | $3.601 \pm 0.042$ | $3.412 \pm 0.058$ | $\mathbf{3.410 \pm 0.027}$ | $5.152 \pm 0.009$ |

