# OpenReview forum: "Hierarchical Rectified Flow Matching with Mini-Batch Couplings"
_ICLR.cc/2026/Conference — Submitted to ICLR 2026_

### Official Review · Reviewer_vg7y · 2025-10-29

**Soundness:** 2
**Presentation:** 3
**Contribution:** 2
**Rating:** 4
**Confidence:** 3

**Summary:**

The paper proposes Hierarchical Flow Matchining with mini-batch coupling, extending the existing HRF method. The coupling can be conducted in two levels: data distribution and velocity distribution, which is shown to gradually adjust the complexity of the distributions across
different levels of the hierarchy via mini-batch couplings.
Extensive numerical experiments are conducted to verify the efficiency.

**Strengths:**

The paper extends the exising HRF framework by introducing coupling, showing that the velocity distribution learnt from coupling data can recover the data distrbution as well. Through some 1D experiments, the authors show that coupling can reduce the multi-modality of velocity distribution, which is convincing. Results of large scale experiments also verify the effectiveness of proposed methods.

**Weaknesses:**

1. The proposed method is a naive combination of two existing methods: HRF and mini-batch coupling, and provides little new insights. It is a well known result that mini-batch coupling can help straighten the velocity and thus address multi-modality in some sense. The authors didn't provide further useful insights for why mini-batch coupling could benefit HRF other than some 1D numerical experiments. Can the authors develop some theories (even toy example is fine) to demonstrate why mini-batching coupling is good for HRF?

2. If I understand correctly, the velocity coupling requires a pretrained velocity model. This limits the practical application of the proposed method, which can only do distillation instead of training from scratch.

3. The authors only compared the proposed algorithm with FM, HRF and FM with coupling, claiming benefits in low NFE regime. The gains in high NFE regime seem quite marginal according to Figure 4(d). In terms of reducing NFE, the proposed method should also compare with other distillation/one-step FM algorithms such as [1,2].

[1] Geng, Zhengyang, et al. "Mean flows for one-step generative modeling." arXiv preprint arXiv:2505.13447 (2025).

[2] Frans, Kevin, et al. "One step diffusion via shortcut models." arXiv preprint arXiv:2410.12557 (2024).

**Questions:**

Please see weakness part.

---

> ### Author Response · Authors · 2025-11-25
> **Response to Reviewer vg7y**
>
> ***QC1: Proposed method is a combination of two existing methods: HRF and mini-batch coupling. It is a well known result that mini-batch coupling can help straighten the velocity and thus address multi-modality in some sense. The authors didn't provide further useful insights for why mini-batch coupling could benefit HRF other than some 1D numerical experiments. Can the authors develop some theories (even toy example is fine) to demonstrate why mini-batching coupling is good for HRF?***
>
> > Our work doesn’t just combine HRF and OT coupling. As stated in L53 and L160, our goal is to reduce the complexity of the target velocity distribution. For this, coupling turns out to be a good method. To our knowledge, hierarchically reducing the target distribution (e.g., velocity, acceleration) has not been studied before.
>
> > Our key insight: couplings at one level simplify the target distribution at the next level of the hierarchy. E.g., data coupling leads to simpler velocity distributions, and velocity coupling simplifies acceleration distributions. This simplification makes learning easier and enables accurate generation. We believe this hierarchical simplification is novel and conceptually valuable, irrespective of the specific technique.
>
> > Further, our method is accompanied by a theoretical analysis. Theorem 3.1 generally characterizes the velocity distribution under an arbitrary joint distribution over data points, which serves as the foundation and inspiration for our coupling strategy. Theorem 3.2 ensures that coupling preserves the marginal distributions over time. Together, these results establish the theoretical validity of our approach.
>
> > In the revised manuscript, we added in Appendix B a theoretical analysis to show that applying mini-batch OT to the 1D source and target data distributions simplifies the velocity distributions. Applying the same argument, we conclude that velocity coupling simplifies the corresponding acceleration distributions.
>
> ***QC2: Velocity coupling requires a pretrained velocity model. This limits the practical application of the proposed method, which can only do distillation instead of training from scratch.***
>
> > Our current velocity coupling approach requires a pretrained velocity model. However, the idea is orthogonal to distillation of models. We think hierarchically simplifying the distributions also allows easier training of distillation models, which results in better generation quality (see results in Appendix G).
>
> ***QC3: The authors only compared the proposed algorithm with FM, HRF and FM with coupling, claiming benefits in a low NFE regime. The gains in a high NFE regime seem quite marginal according to Figure 4(d). In terms of reducing NFE, the proposed method should also compare with other distillation/one-step FM algorithms such as [1,2].***
>
> > Distillation and one-step FM algorithms are orthogonal to our method. We use mini-batch coupling to simplify the distributions that need to be modeled. Distillation methods can be combined with our hierarchical RF models. In fact, we incorporated the shortcut model [1] into HRF2-D&V and show results in Appendix G of the revised manuscript. We observe that combining the data and velocity coupling with the ShortCut model consistently improves upon the ShortCut model. The following table shows the FID comparison on CIFAR-10 dataset for ShortCut model, HRF2-D&V model, and the combined HRF2-D&V with ShortCut model. This confirms that data and velocity coupling and shortcut-based distillation can be jointly used within our framework.
>
> | NFE | ShortCut | HRF2-D&V | HRF2-D&V+ShortCut |
> |--|--|--|--|
> | 1 | 41.60 | 15.17 | 16.95 |
> | 4 | 15.78 | 6.78 | 11.35 |
> | 8 | 12.68 | 6.06 | 10.45 |
>
> References:
> [1] Frans et al. One Step Diffusion via Shortcut Models. ICLR, 2025

---

### Official Review · Reviewer_2cZT · 2025-10-31

**Soundness:** 2
**Presentation:** 3
**Contribution:** 1
**Rating:** 4
**Confidence:** 4

**Summary:**

This paper mainly focuses on gradually simplifying the complexity of the distributions across hierarchy in hierarchical flow matching using mini-batch optimal transport (OT) coupling. The paper empirically compares different schemes of coupling, including data coupling and joint data-velocity coupling. The paper experiments on synthetic and image datasets to show the effectiveness of the proposed method.

**Strengths:**

•	The paper is well-written and clearly-organized.

•	The empirical discovery that using mini-batch OT coupling in the current hierarchy level simplifies the distribution at the next level is interesting and could be considered in other scenarios of generative modeling like diffusion models.

**Weaknesses:**

•	The novelty of the paper is limited and incremental, the usage of minibatch OT coupling in flow matching is already proposed in [1]. The main point of the paper that using mini-batch OT inherently simplifies the velocity distribution is only intuitively explained without further theoretical mathematical evaluation.

•	The paper lacks further theoretical discussion and comparison between “data coupling” and “joint data and velocity coupling”.

•	The method is not simulation-free, which increases the training cost.

•	[1] Aram-Alexandre Pooladian, Heli Ben-Hamu, Carles Domingo-Enrich, Brandon Amos, Yaron Lipman, and Ricky TQ Chen. Multisample flow matching: Straightening flows with minibatch couplings. In Proc. ICML, 2023

**Questions:**

•	As shown in Figure 3(d), HRF2-D and HRF2-D&V performs nearly the same, with HRF2-D&V slightly better. In contrast, HRF2-D&V performs worse than HRF2-D when NFE is large on the real image datasets. How to choose between “data coupling” and “joint data and velocity coupling” given a certain task and a certain NFE?

•	Could the authors provide a theoretical validation of “minibatch OT coupling inherently simplifies the velocity distribution on the next hierarchy” on a simple distribution like Gaussian distribution?

---

> ### Author Response · Authors · 2025-11-25
> **Response to Reviewer 2cZT**
>
> ***QB1: Novelty of the paper: minibatch OT coupling in flow matching is already proposed in [1]. The main point of the paper: using mini-batch OT simplifies the velocity distribution is only intuitively explained without further theoretical mathematical evaluation.***
>
> > Our work doesn’t just combine HRF and OT coupling. As stated in L53 and L160, our goal is to reduce the complexity of the target velocity distribution. For this, coupling turns out to be a good method. To our knowledge, hierarchically reducing the target distribution (e.g., velocity, acceleration) has not been studied before.
>
> > Our key insight: couplings at one level simplify the target distribution at the next level of the hierarchy. E.g., data coupling leads to simpler velocity distributions, and velocity coupling simplifies acceleration distributions. This simplification makes learning easier and enables accurate generation. We believe this hierarchical simplification is novel and conceptually valuable, irrespective of the specific technique.
>
> > Further, our method is accompanied by a theoretical analysis. Theorem 3.1 generally characterizes the velocity distribution under an arbitrary joint distribution over data points, which serves as the foundation and inspiration for our coupling strategy. Theorem 3.2 ensures that coupling preserves the marginal distributions over time. Together, these results establish the theoretical validity of our approach.
>
> > In the revised manuscript, we added in Appendix B a theoretical analysis to show that applying mini-batch OT to the 1D source and target data distributions simplifies the velocity distributions. Applying the same argument, we conclude that velocity coupling simplifies the corresponding acceleration distributions.
>
> ***QB2: The paper lacks further theoretical discussion and comparison between “data coupling” and “joint data and velocity coupling”.***
>
> > We added a theory section in Appendices B and C, justifying how mini-batch OT coupling reduces multi-modality in a 1d Gaussian mixture case. The theorem and corollary explicitly show that for data coupling, the multi-modality of the velocity distribution is reduced. The result is also consistent such that multi-modality vanishes when the batch size goes to infinity. The joint coupling case is a natural extension of the theorem. This serves as an illustration of how in theory coupling helps us to reduce multi-modality in the next hierarchy level.
>
> ***QB3: The method is not simulation-free, which increases the training cost.***
>
> >  We discussed the computational costs in Appendix D and reported the training time and memory usage in Tables 8-11 and the analysis of OT cost in Table 12. The training time for HRF2-D is nearly identical to HRF2 because mini-batch OT computation overhead is relatively small (<2%) compared to the gradient evaluation. For HRF2-D&V, we require $(v_0,v_1)$-pairs, which need to be generated via model inference. Therefore, we generate $(v_0, v_1)$-pairs first and train the HRF-D&V model afterwards. Velocity sample generation can be trivially parallelized across the number of available GPUs. Once the data is generated, training speed and memory usage are comparable to other baselines.
>
>
> ***QB4: As shown in Figure 3(d), HRF2-D and HRF2-D&V perform nearly the same, with HRF2-D&V slightly better. In contrast, HRF2-D&V performs worse than HRF2-D when NFE is large on the real image datasets. How to choose between “data coupling” and “joint data and velocity coupling” given a certain task and a certain NFE?***
>
> > In Figure 3(d), the gap between HRF2-D and HRF2-D&V appears mainly at low NFE. Empirically, we observe that, except for simple 1D data, the advantage of joint data and velocity coupling is most apparent at low NFE. When NFE is large, the advantage diminishes and can slightly degrade performance, so we recommend HRF2-D&V when low inference cost is required and HRF2-D when more steps are possible.
>
> ***QB5: Could the authors provide a theoretical validation of “minibatch OT coupling inherently simplifies the velocity distribution on the next hierarchy” on a simple distribution like Gaussian distribution?***
>
> > Thanks for the suggestion. We have included the additional theoretical analysis in Appendix B and C.

---

### Official Review · Reviewer_NdTd · 2025-11-04

**Soundness:** 3
**Presentation:** 2
**Contribution:** 2
**Rating:** 2
**Confidence:** 3

**Summary:**

The paper extends the hierarchical rectified flow matching models by incorporating mini-batch optimal transport couplings between the source and target samples for the data and velocity. It is shown through theory and empirical evidence that the mini-batch coupling (for a large batch size) simplifies the distribution of velocity close to t=0 (closer to the source distribution) for rectified flow models. This observation motivated the authors to apply mini-batch OT in the two-level hierarchical rectified flow model (HRF2). The paper shows improvement in FID with a few NFEs for popular image benchmarks and synthetic datasets.

**Strengths:**

I like the presentation of the approach. The use of mini-batch OT in the two levels of hierarchical rectified flow is intuitive and well-motivated from experiments on bi-modal Gaussian data. The theoretical results are interesting and easy to follow. The experiments do show improvement in lower FID compared to baselines with low NFEs.

**Weaknesses:**

Major:

Novelty - The paper's main contribution is to use mini-batch OT for training the HRF2 model. The mini-batch OT has been previously used for CFM models (OT-CFM). I understand that this paper extends this to HRF models, but in my view, this is a very limited novelty.

Training cost - Flow matching training with the mini-batch OT is expensive. This method adds computing another OT map for the velocity distribution. In addition, HRF2 with velocity coupling requires a simulation from ODE flow for the velocity distribution. I would request the authors to add the training cost analysis of their method.

Other:

Suboptimal baselines - The OT-CFM results on CIFAR10 seem to have FID > 5 for NFE 100. However, Tong et al. (OT-CFM), Guo and Schwing (VRFM), and Samaddar et al. (Latent-CFM) all reported FID < 5 for NFE 100. This suggests that the baseline training or inference was suboptimal.

**Questions:**

1. In Fig. 1(a), after 100 steps of ODE steps, why does the velocity distribution not reach a univariate Gaussian (as shown in Fig. 2 in the original HRF paper, Zhang et al. 2025)?

2. Please enlarge the legends in the 2d sample plots, Fig. 3, top panel (two plots on the right).

3. Please report the overall performance of all the competing approaches at the final Euler step for the image benchmarks.

4. Can the author comment on the use of adaptive solvers instead of fixed-step Euler?

---

> ### Author Response · Authors · 2025-11-25
> **Response to Reviewer NdTd**
>
> ***QA1: Novelty - Main contribution is to use mini-batch OT for training HRF2. Mini-batch OT has been used for CFM models (OT-CFM). This paper extends this to HRF.***
>
> > Our work doesn’t just combine HRF and OT coupling. As stated in L53 and L160, our goal is to reduce the complexity of the target velocity distribution. For this, coupling turns out to be a good method. To our knowledge, hierarchically reducing the target distribution (e.g., velocity, acceleration) has not been studied before.
>
> > Our key insight: couplings at one level simplify the target distribution at the next level of the hierarchy. E.g., data coupling leads to simpler velocity distributions, and velocity coupling simplifies acceleration distributions. This simplification makes learning easier and enables accurate generation. We believe this hierarchical simplification is novel and conceptually valuable, irrespective of the specific technique.
>
> > Further, our method is accompanied by a theoretical analysis. Theorem 3.1 generally characterizes the velocity distribution under an arbitrary joint distribution over data points, which serves as the foundation and inspiration for our coupling strategy. Theorem 3.2 ensures that coupling preserves the marginal distributions over time. Together, these results establish the theoretical validity of our approach.
>
> > In the revised manuscript, we added in Appendix B a theoretical analysis to show that applying mini-batch OT to the 1D source and target data distributions simplifies the velocity distributions. Applying the same argument, we conclude that velocity coupling simplifies the corresponding acceleration distributions.
>
> ***QA2: Training cost - FM training with mini-batch OT is expensive. This method adds another OT map. Further, HRF2 with velocity coupling requires a simulation. Training cost analysis?***
>
> > We discussed computational cost in Appendix D, reported training time and memory usage in Tables 8-11, and the analysis of OT cost in Table 12. The training time for HRF2-D is nearly identical to HRF2 because mini-batch OT computation overhead is relatively small (<2%) compared to the gradient evaluation. For HRF2-D&V, we require $(v_0,v_1)$-pairs, which need to be generated via model inference. Hence, we generate $(v_0, v_1)$-pairs first and train the HRF-D&V model afterwards. Velocity sample generation can be trivially parallelized across the number of available GPUs. Once the data is generated, training speed and memory usage are comparable to other baselines.
>
> ***QA3: Suboptimal baselines - OT-CFM results on CIFAR10 seem to have FID > 5 for NFE 100. Others reported FID < 5 for NFE 100.***
>
> > In the OT-CFM paper, the reported FIDs for I-CFM (corresponding to RF in our paper) and OT-CFM at NFE = 100 are 4.461 and 4.443, respectively. Our reproduced and reported results are 4.588 and 4.952 as shown in Table 2 in Appendix C.2, which we consider a reasonable reproduction. We used the same U-Net architecture as OT-CFM, while VRFM and Latent-CFM adopt different model architectures, so we do not make a direct comparison with them. Even when using the reported OT-CFM values, our method still achieves a lower FID of 4.301.
>
> ***QA4: Fig. 1(a), after 100 steps of ODE integration, why does the velocity distribution not reach a univariate Gaussian (as in Fig. 2 of HRF)?***
>
> > The velocity distribution will not reach a univariate Gaussian. It will be a shifted data distribution at t=0. Just as shown in Fig. 2(a) of the original HRF paper, our velocity in Fig 1(a) is a shifted Gaussian mixture.
>
> ***QA5: Enlarge legends in 2d data plots.***
>
> > Revised.
>
> ***QA6: Report results of all baselines at final Euler step for image benchmarks.***
>
> > We already reported them in Table 1-3 in Appendix C.2.
>
> ***QA7: Use of adaptive solvers instead of fixed-step Euler?***
>
> > Great suggestion. Our sampler has two nested integrations. The inner integration is a standard flow matching update. This inner integration can be replaced by any adaptive solver without changing the formulation. We focused on fixed step Euler for clarity. Here is a comparison between fixed Euler and adaptive sampling (dopri5) on CIFAR-10 data. More details are shown in Appendix H.
>
> | Total NFEs | RF (35.75M) | OT-CFM (35.75M) | HRF2 (44.81M) | HRF2-D (44.81M) | HRF2-D&V (44.81M) |
> |--|--|--|--|--|--|
> | 100 | 4.588 ± 0.013 | 4.952 ± 0.012 | 4.334 ± 0.054 | **4.301 ± 0.022** | 5.078 ± 0.044 |
> | 500 | 3.887 ± 0.035 | 4.184 ± 0.086 | 3.706 ± 0.043 | **3.578 ± 0.028** | 5.095 ± 0.032 |
> | Adaptive | 3.688 ± 0.077 | 3.601 ± 0.042 | 3.412 ± 0.058 | **3.410 ± 0.027** | 5.152 ± 0.009 |

---

### Author Response · Authors · 2025-11-25

We appreciate all reviewers’ comments and suggestions. We have revised the paper based on their feedback and respond to each reviewer below. In the revised manuscript, we included in Appendix B a theoretical analysis which shows that using mini-batch OT coupling for data simplifies the velocity distributions. In Appendix C, we detail the acceleration distributions and show that mini-batch OT coupling on the velocity simplifies acceleration distributions. We included additional experimental results to show that other few step generation techniques can be incorporated in our framework (see Appendix G). Moreover, since the distributions get less multimodal, it gets easier to train shortcut models, which results in better generation quality. The changes are highlighted in blue.

---

### Author Response · Authors · 2025-12-03
**Final Remarks**

We thank all reviewers for their valuable questions and thoughtful feedback. They consistently described the method as “intuitive and well motivated” (NdTd), and noted that the theoretical results are “interesting and easy to follow” (NdTd, 2cZT). Reviewers also highlighted that the paper is “well-written” (2cZT) and that the 1D experiments make the theory “convincing” (vg7y). Across evaluations, reviewers agreed that the experiments “show improvement in lower FID compared to baselines with low NFEs” (NdTd) and that the large-scale results “verify the effectiveness” (vg7y) of the proposed methods.

Regarding the question on novelty, contributions, and theoretical results, we’d like to note:

1) **Novel idea** – We present a hierarchical framework to simplify target distributions, which to our knowledge has not been explored in prior diffusion models.

2) **New theoretical results** – (A) A general characterization of the velocity distribution under an arbitrary joint distribution over data points (Theorem 3.1); (B) A proof that hierarchical rectified flow with mini-batch coupling preserves marginals over time (Theorem 3.2); (C) A theoretical analysis for 1D data, showing that mini batch OT simplifies velocity distributions, and, following the same reasoning, that velocity coupling simplifies accelerations (Appendix B and Appendix C).

3) **Practical relevance** – HRF2-D outperforms OT-CFM and other baselines for most NFEs on MNIST, CIFAR-10, and CelebA-HQ; HRF2-D&V yields further gains for few-step generation.

4) **Compatibility** – HRF2 with data and velocity coupling is orthogonal to distillation. Incorporating the ShortCut model into HRF2-D&V (Appendix G) shows that data and velocity coupling consistently improve the distilled model.

We have carefully answered all questions and we have incorporated all additional results, clarifications, and theoretical discussion into the revised manuscript. All suggested changes will appear in the final version.

In summary, our work shows both theoretical advances and practical gains, and we hope to inspire future research on hierarchically simplifying target distributions in generative modeling.

---

### Meta-Review · Area_Chair_dnee · 2026-01-11

**Summary:**

# Summary of concerns


### Common concerns: Reviewer NdTd,  Reviewer 2cZT, Reviewer vg7y
1. The paper has limited novelty, and it's a simple combination of mini-batch OT and HRF2
    * **Author replies**: The authors claim that the goal of the paper is to reduce the
    complexity of the velocity distribution, for which mini-Batch couplings turn out to be
    a good solution. The paper also provides a theoretical analysis.
    * **AC comment**: I think this remains an outstanding concern.
    While the motivation of the paper is reasonable and can be presented differently,
    the paper itself is a combination of hierarchical rectified flow matching with mini-batch coupling,
    as pointed out by all reviewers. The original Multisample Flow Matching has already motivated
    the usage of Minibatch Couplings to generate straight paths and thus reduce multi-modality.
    The extra theoretical analysis is interesting, but it is done on a rather simple data space,
    which also limits the scope and applicability.


### Reviewer NdTd
1. Training cost of mini-batch OT is expensive:
    * **Author replies**: The authors add an analysis of the runtime and show that the cost of
    mini-batch OT is relatively small compared to gradient evaluation. Velocity sample generation
    can be parallelized across GPUs.
    * **AC comment**: I think the concern on mini-batch OT is well addressed. However, the cost
    of velocity generation is still a concern. While it's possible to do it in parallel, it will definitely require additional computing resources. It's needed to report how much compute and time it takes.

### Reviewer 2cZT
1. The method is not simulation-free, which increases the training cost.
    * See Reviewer NdTd 1 above.


### Reviewer vg7y
1. Velocity coupling requires a pretrained velocity model. This limits the practical application of the proposed method.
    * **Author replies**: The authors acknowledge the limitation. The authors argue that hierarchically simplifying the distributions also allows easier training of distillation models.
    * **AC comment**:  I think the concern is well addressed. HRF2 with velocity coupling can build on
    top of the HRF2 data coupling model, and the whole pipeline can still be trained from scratch, so it
    is still practical.
2. Needs to add comparison to few-step distillation methods.
    * **Author replies**: Distillation is orthogonal to the paper, which aims to use mini-batch coupling to simplify the distributions and thus improve few-step generations. The authors also add experiments to show that the proposed
    method can be combined with distillation methods such as ShortCut to improve its performance.
    * **AC comment**: I think the concern is well addressed.

**Reviewer Concerns:**

Overall, I think the major concern of limited technical novelty on the paper remains outstanding as discussed above.

**Reviewer Scores:**

Reviewer NdTd: maintain 2

Reviewer 2cZT: maintain 4

Reviewer vg7y: maintain 4

---

### Decision · Program_Chairs · 2026-01-26

Reject